# Limitations of principal components in quantitative genetic association models for human studies

**Yiqi Yao[1†], Alejandro Ochoa[1,2]\***

[1]Department of Biostatistics and Bioinformatics, Duke University, Durham, United States; [2]Duke Center for Statistical Genetics and Genomics, Duke University, Durham, United States

**Abstract** Principal Component Analysis (PCA) and the Linear Mixed-effects Model (LMM), sometimes in combination, are the most common genetic association models. Previous PCA-LMM comparisons give mixed results, unclear guidance, and have several limitations, including not varying the number of principal components (PCs), simulating simple population structures, and inconsistent use of real data and power evaluations. We evaluate PCA and LMM both varying number of PCs in realistic genotype and complex trait simulations including admixed families, subpopulation trees, and real multiethnic human datasets with simulated traits. We find that LMM without PCs usually performs best, with the largest effects in family simulations and real human datasets and traits without environment effects. Poor PCA performance on human datasets is driven by large numbers of distant relatives more than the smaller number of closer relatives. While PCA was known to fail on family data, we report strong effects of family relatedness in genetically diverse human datasets, not avoided by pruning close relatives. Environment effects driven by geography and ethnicity are better modeled with LMM including those labels instead of PCs. This work better characterizes the severe limitations of PCA compared to LMM in modeling the complex relatedness structures of multiethnic human data for association studies.

**\*For correspondence:**
alejandro.ochoa@duke.edu

**Present address:** [†]BenHealth Consulting, Shanghai, Shanghai, China

## Editor's evaluation

This is an important paper that presents compelling arguments (based on simulation and comprehensively reviewed background theory) that Linear Mixed Models generally should perform better at correcting for genetic and environmental confounding in GWAS than more commonly used Principal Components methods.

## Introduction

The goal of a genetic association study is to identify loci whose genotype variation is significantly correlated to given trait. Naive association tests assume that genotypes are drawn independently from a common allele frequency. This assumption does not hold for structured populations, which includes multiethnic cohorts and admixed individuals (ancient relatedness), and for family data (recent relatedness; *Astle and Balding, 2009*). Association studies of admixed and multiethnic cohorts, the focus of this work, are becoming more common, are believed to be more powerful, and are necessary to bring more equity to genetic medicine (*Rosenberg et al., 2010*; *Hoffman and Dubé, 2013*; *Coram et al., 2013*; *Medina-Gomez et al., 2015*; *Conomos et al., 2016a*; *Hodonsky et al., 2017*; *Martin et al., 2017a*; *Martin et al., 2017b*; *Hindorff et al., 2018*; *Hoffmann et al., 2018*; *Mogil et al., 2018*; *Roselli et al., 2018*; *Wojcik et al., 2019*; *Peterson et al., 2019*; *Zhong et al., 2019*; *Hu*

*et al., 2020*; *Simonin-Wilmer et al., 2021*; *Kamariza et al., 2021*; *Lin et al., 2021*; *Mahajan et al., 2022*; *Hou et al., 2023a*). When insufficient approaches are applied to data with relatedness, their association statistics are miscalibrated, resulting in excess false positives and loss of power (*Devlin and Roeder, 1999*; *Voight and Pritchard, 2005*; *Astle and Balding, 2009*). Therefore, many specialized approaches have been developed for genetic association under relatedness, of which PCA and LMM are the most popular.

Genetic association with PCA consists of including the top eigenvectors of the population kinship matrix as covariates in a generalized linear model (*Zhang et al., 2003*; *Price et al., 2006*; *Bouaziz et al., 2011*). These top eigenvectors are a new set of coordinates for individuals that are commonly referred to as PCs in genetics (*Patterson et al., 2006*), the convention adopted here, but in other fields PCs instead denote what in genetics would be the projections of loci onto eigenvectors, which are new independent coordinates for loci (*Jolliffe, 2002*). The direct ancestor of PCA association is structured association, in which inferred ancestry (genetic cluster membership, often corresponding with labels such as "European", "African", "Asian", etc.) or admixture proportions of these ancestries are used as regression covariates (*Pritchard et al., 2000*). These models are deeply connected because PCs map to ancestry empirically (*Alexander et al., 2009*; *Zhou et al., 2016*) and theoretically (*McVean, 2009*; *Zheng and Weir, 2016*; *Cabreros and Storey, 2019*; *Chiu et al., 2022*), and they work as well as global ancestry in association studies but are estimated more easily (*Patterson et al., 2006*; *Zhao et al., 2007*; *Alexander et al., 2009*; *Bouaziz et al., 2011*). Another approach closely related to PCA is nonmetric multidimensional scaling (*Zhu and Yu, 2009*). PCs are also proposed for modeling environment effects that are correlated to ancestry, for example, through geography (*Novembre et al., 2008*; *Zhang and Pan, 2015*; *Lin et al., 2021*). The strength of PCA is its simplicity, which as covariates can be readily included in more complex models, such as haplotype association (*Xu and Guan, 2014*) and polygenic models (*Qian et al., 2020*). However, PCA assumes that the underlying relatedness space is low dimensional (or low rank), so it can be well modeled with a small number of PCs, which may limit its applicability. PCA is known to be inadequate for family data (*Patterson et al., 2006*; *Zhu and Yu, 2009*; *Thornton and McPeek, 2010*; *Price et al., 2010*), which is called 'cryptic relatedness' when it is unknown to the researchers, but no other troublesome cases have been confidently identified. Recent work has focused on developing more scalable versions of the PCA algorithm (*Lee et al., 2012*; *Abraham and Inouye, 2014*; *Galinsky et al., 2016*; *Abraham et al., 2017*; *Agrawal et al., 2020*). PCA remains a popular and powerful approach for association studies.

The other dominant association model under relatedness is the LMM, which includes a random effect parameterized by the kinship matrix. Unlike PCA, LMM does not assume that relatedness is low-dimensional, and explicitly models families via the kinship matrix. Early LMMs used kinship matrices estimated from known pedigrees or using methods that captured recent relatedness only, and modeled population structure (ancestry) as fixed effects (*Yu et al., 2006*; *Zhao et al., 2007*; *Zhu and Yu, 2009*). Modern LMMs estimate kinship from genotypes using a non-parametric estimator, often referred to as a genetic relationship matrix, that captures the combined covariance due to family relatedness and ancestry (*Kang et al., 2008*; *Astle and Balding, 2009*; *Ochoa and Storey, 2021*). Like PCA, LMM has also been proposed for modeling environment correlated to genetics (*Vilhjálmsson and Nordborg, 2013*; *Wang et al., 2022*). The classic LMM assumes a quantitative (continuous) complex trait, the focus of our work. Although case-control (binary) traits and their underlying ascertainment are theoretically a challenge (*Yang et al., 2014*), LMMs have been applied successfully to balanced case-control studies (*Astle and Balding, 2009*; *Kang et al., 2010*) and simulations (*Price et al., 2010*; *Wu et al., 2011*; *Sul and Eskin, 2013*), and have been adapted for unbalanced case-control studies (*Zhou et al., 2018*). However, LMMs tend to be considerably slower than PCA and other models, so much effort has focused on improving their runtime and scalability (*Aulchenko et al., 2007*; *Kang et al., 2008*; *Kang et al., 2010*; *Zhang et al., 2010*; *Lippert et al., 2011*; *Yang et al., 2011*; *Listgarten et al., 2012*; *Zhou and Stephens, 2012*; *Svishcheva et al., 2012*; *Loh et al., 2015*; *Zhou et al., 2018*).

An LMM variant that incorporates PCs as fixed covariates is tested thoroughly in our work. Since PCs are the top eigenvectors of the same kinship matrix estimate used in modern LMMs (*Astle and Balding, 2009*; *Janss et al., 2012*; *Hoffman and Dubé, 2013*; *Zhang and Pan, 2015*), then population structure is modeled twice in an LMM with PCs. However, some previous work has found the

**Table 1.** Previous PCA-LMM evaluations in the literature.

| Publication | Sim. Genotypes | | | General | | | PCs($r$) | Best |
|---|---|---|---|---|---|---|---|---|
| | Type* | $K^\dagger$ | $F_{ST}^\ddagger$ | Real $^\S$ | Trait $^\P$ | Power | | |
| *Zhao et al., 2007* | | | | ✓ | Q | ✓ | 8 | LMM |
| *Zhu and Yu, 2009* | I, A, F | 3, 8 | ≤0.15 | ✓ | Q | ✓ | 1–22 | LMM |
| *Astle and Balding, 2009* | I | 3 | 0.10 | | CC | ✓ | 10 | Tie |
| *Kang et al., 2010* | | | | ✓ | Both | | 2–100 | LMM |
| *Price et al., 2010* | I, F | 2 | 0.01 | | CC | | 1 | Mixed |
| *Wu et al., 2011* | I, A | 2–4 | 0.01 | | CC | ✓ | 10 | Mixed |
| *Liu et al., 2011* | S, A | 2–3 | R | | Q | ✓ | 10 | Tie |
| *Sul and Eskin, 2013* | I | 2 | 0.01 | | CC | | 1 | Tie |
| *Tucker et al., 2014* | I | 2 | 0.05 | ✓ | Both | ✓ | 5 | Tie |
| *Yang et al., 2014* | | | | ✓ | CC | ✓ | 5 | Tie |
| *Song et al., 2015* | S, A | 2–3 | R | | Q | | 3 | LMM |
| *Loh et al., 2015* | | | | ✓ | Q | ✓ | 10 | LMM |
| *Zhang and Pan, 2015* | | | | ✓ | Q | ✓ | 20–100 | LMM |
| *Liu et al., 2016* | | | | ✓ | Q | ✓ | 3–6 | LMM |
| *Sul et al., 2018* | | | | ✓ | Q | | 100 | LMM |
| *Loh et al., 2018* | | | | ✓ | Both | ✓ | 20 | LMM |
| *Mbatchou et al., 2021* | | | | ✓ | Both | | 1 | LMM |
| This work | A, T, F | 10–243 | ≤0.25 | ✓ | Q | ✓ | 0–90 | LMM |

*Genotype simulation types. I: Independent subpopulations; S: subpopulations (with parameters drawn from real data); A: Admixture; T: Subpopulation Tree; F: Family.

$^\dagger$Model dimension (number of subpopulations or ancestries).

$^\ddagger$R: simulated parameters based on real data, $F_{ST}$ not reported.

$^\S$Evaluations using unmodified real genotypes.

$^\P$Q: quantitative; CC: case-control.

apparent redundancy of an LMM with PCs beneficial (*Price et al., 2010*; *Tucker et al., 2014*; *Zhang and Pan, 2015*), while others did not (*Liu et al., 2011*; *Janss et al., 2012*), and the approach continues to be used (*Zeng et al., 2018*; *Mbatchou et al., 2021*), although not always (*Matoba et al., 2020*). Recall that early LMMs used kinship to model family relatedness only, so population structure had to be modeled separately in those models, in practice as admixture fractions instead of PCs (*Yu et al., 2006*; *Zhao et al., 2007*; *Zhu and Yu, 2009*). The LMM with PCs (vs no PCs) is also believed to help better model loci that have experienced selection (*Price et al., 2010*; *Vilhjálmsson and Nordborg, 2013*) and environment effects correlated with genetics (*Zhang and Pan, 2015*).

LMM and PCA are closely related models (*Astle and Balding, 2009*; *Janss et al., 2012*; *Hoffman and Dubé, 2013*; *Zhang and Pan, 2015*), so similar performance is expected particularly under low-dimensional relatedness. Direct comparisons have yielded mixed results, with several studies finding superior performance for LMM, notably from papers promoting advances in LMMs, while many others report comparable performance (*Table 1*). No papers find that PCA outperforms LMM decisively, although PCA occasionally performs better in isolated and artificial cases or individual measures, often with unknown significance. Previous studies generally used either only simulated or only real genotypes, with only two studies using both. The simulated genotype studies, which tended to have low model dimensions and $F_{ST}$, were more likely to report ties or mixed results (6/8), whereas real genotypes tended to clearly favor LMMs (9/11). Similarly, 10/12 papers with quantitative traits favor LMMs, whereas 6/9 papers with case-control traits gave ties or mixed results—the only factor we do not explore in this work. Additionally, although all previous evaluations measured type I error (or proxies such as genomic

inflation factors *Devlin and Roeder, 1999* or QQ plots), a large fraction (6/17) did not measure power (or proxies such as ROC curves), and only four used more than one number of PCs for PCA. Lastly, no consensus has emerged as to why LMM might outperform PCA or vice versa (*Price et al., 2010*; *Sul and Eskin, 2013*; *Price et al., 2013*; *Hoffman and Dubé, 2013*), or which features of the real datasets are critical for the LMM advantage other than family relatedness, resulting in unclear guidance for using PCA. Hence, our work includes real and simulated genotypes with higher model dimensions and $F_{\text{ST}}$ matching that of multiethnic human cohorts (*Ochoa and Storey, 2021*; *Ochoa and Storey, 2019*), we vary the number of PCs, and measure robust proxies for type I error control and calibrated power.

In this work, we evaluate the PCA and LMM association models under various numbers of PCs, which are included in LMMs too. We use genotype simulations (admixture, family, and subpopulation tree models) and three real datasets: the 1000 Genomes Project (*Abecasis et al., 2010*; *Abecasis et al., 2012*), the Human Genome Diversity Panel (HGDP) (*Cann et al., 2002*; *Rosenberg et al., 2002*; *Bergström et al., 2020*), and Human Origins (*Patterson et al., 2012*; *Lazaridis et al., 2014*; *Lazaridis et al., 2016*; *Skoglund et al., 2016*). We simulate quantitative traits from two models: fixed effect sizes (FES) construct coefficients inverse to allele frequency, which matches real data (*Park et al., 2011*; *Zeng et al., 2018*; *O'Connor et al., 2019*) and corresponds to high pleiotropy and strong balancing selection (*Simons et al., 2018*) and strong negative selection (*Zeng et al., 2018*; *O'Connor et al., 2019*), which are appropriate assumptions for diseases; and random coefficients (RC), which are drawn independent of allele frequency, and corresponds to neutral traits (*Zeng et al., 2018*; *Simons et al., 2018*). LMM without PCs consistently performs best in simulations without environment, and greatly outperforms PCA in the family simulation and in all real datasets. The tree simulations, which model subpopulations with the tree but exclude family structure, do not recapitulate the real data results, suggesting that family relatedness in real data is the reason for poor PCA performance. Lastly, removing up to 4th degree relatives in the real datasets recapitulates poor PCA performance, showing that the more numerous distant relatives explain the result, and suggesting that PCA is generally not an appropriate model for real data. We find that both LMM and PCA are able to model environment effects correlated with genetics, and LMM with PCs gains a small advantage in this setting only, but direct modeling of environment performs much better. All together, we find that LMMs without PCs are generally a preferable association model, and present novel simulation and evaluation approaches to measure the performance of these and other genetic association approaches.

**Table 2.** Features of simulated and real human genotype datasets.

| Dataset | Type | Loci($m$) | Ind. ($n$) | Subpops.* ($K$) | Causal loci[†] ($m_1$) | $F_{\text{ST}}$[‡] |
|---|---|---|---|---|---|---|
| Admix. Large sim. | Admix. | 100 000 | 1000 | 10 | 100 | 0.1 |
| Admix. Small sim. | Admix. | 100 000 | 100 | 10 | 10 | 0.1 |
| Admix. Family sim. | Admix.+Pedig. | 100 000 | 1000 | 10 | 100 | 0.1 |
| Human Origins | Real | 190 394 | 2922 | 11–243 | 292 | 0.28 |
| HGDP | Real | 771 322 | 929 | 7–54 | 93 | 0.28 |
| 1000 Genomes | Real | 1 111 266 | 2504 | 5–26 | 250 | 0.22 |
| Human Origins sim. | Tree | 190 394 | 2922 | 243 | 292 | 0.23 |
| HGDP sim. | Tree | 771 322 | 929 | 54 | 93 | 0.25 |
| 1000 Genomes sim. | Tree | 1 111 266 | 2504 | 26 | 250 | 0.21 |

Ind.($n$)

*For admixed family, ignores additional model dimension of 20 generation pedigree structure. For real datasets, lower range is continental subpopulations, upper range is number of fine-grained subpopulations.

[†]$m_1 = \text{round}(nh^2/8)$ to balance power across datasets, shown for $h^2 = 0.8$ only.

[‡]Model parameter for simulations, estimated value on real datasets.

# Results

## Overview of evaluations

We use three real genotype datasets and simulated genotypes from six population structure scenarios to cover various features of interest (*Table 2*). We introduce them in sets of three, as they appear in the rest of our results. Population kinship matrices, which combine population and family relatedness, are estimated without bias using popkin (*Ochoa and Storey, 2021*; *Figure 1*). The first set of three simulated genotypes are based on an admixture model with 10 ancestries (*Figure 1A*; *Ochoa and Storey, 2021*; *Gopalan et al., 2016*; *Cabreros and Storey, 2019*). The 'large' version (1000 individuals) illustrates asymptotic performance, while the 'small' simulation (100 individuals) illustrates model overfitting. The 'family' simulation has admixed founders and draws a 20-generation random pedigree with assortative mating, resulting in a complex joint family and ancestry structure in the last generation (*Figure 1B*). The second set of three are the real human datasets representing global human diversity: Human Origins (*Figure 1D*), HGDP (*Figure 1G*), and 1000 Genomes (*Figure 1J*), which are enriched for small minor allele frequencies even after MAF <1% filter (*Figure 1C*). Last are subpopulation tree simulations (*Figure 1F, I, L*) fit to the kinship (*Figure 1E, H and K*) and MAF (*Figure 1C*) of each real human dataset, which by design do not have family structure.

All traits in this work are simulated. We repeated all evaluations on two additive quantitative trait models, *fixed effect sizes* (FES) and *random coefficients* (RC), which differ in how causal coefficients are constructed. The FES model captures the rough inverse relationship between coefficient and minor allele frequency that arises under strong negative and balancing selection and has been observed in numerous diseases and other traits (*Park et al., 2011*; *Zeng et al., 2018*; *Simons et al., 2018*; *O'Connor et al., 2019*), so it is the focus of our results. The RC model draws coefficients independent of allele frequency, corresponding to neutral traits (*Zeng et al., 2018*; *Simons et al., 2018*), which results in a wider effect size distribution that reduces association power and effective polygenicity compared to FES.

We evaluate using two complementary measures: (1) $\mathrm{SRMSD}_p$ (p-value signed root mean square deviation) measures p-value calibration (closer to zero is better), and (2) $\mathrm{AUC}_{\mathrm{PR}}$ (precision-recall area under the curve) measures causal locus classification performance (higher is better; *Figure 2*). $\mathrm{SRMSD}_p$ is a more robust alternative to the common inflation factor $\lambda$ and type I error control measures; there is a correspondence between $\lambda$ and $\mathrm{SRMSD}_p$, with $\mathrm{SRMSD}_p > 0.01$ giving $\lambda > 1.06$ (*Figure 2—figure supplement 1*) and thus evidence of miscalibration close to the rule of thumb of $\lambda > 1.05$ (*Price et al., 2010*). There is also a monotonic correspondence between $\mathrm{SRMSD}_p$ and type I error rate (*Figure 2—figure supplement 2*). $\mathrm{AUC}_{\mathrm{PR}}$ has been used to evaluate association models (*Rakitsch et al., 2013*), and reflects calibrated statistical power (*Figure 2—figure supplement 3*) while being robust to miscalibrated models (Appendix 2).

Both PCA and LMM are evaluated in each replicate dataset including a number of PCs $r$ between 0 and 90 as fixed covariates. In terms of p-value calibration, for PCA the best number of PCs $r$ (minimizing mean $|\mathrm{SRMSD}_p|$ over replicates) is typically large across all datasets (*Table 3*), although much smaller $r$ values often performed as well (shown in following sections). Most cases have a mean $|\mathrm{SRMSD}_p| < 0.01$, whose p-values are effectively calibrated. However, PCA is often miscalibrated on the family simulation and real datasets (*Table 3*). In contrast, for LMM, $r = 0$ (no PCs) is always best, and is always calibrated. Comparing LMM with $r = 0$ to PCA with its best $r$, LMM always has significantly smaller $|\mathrm{SRMSD}_p|$ than PCA or is statistically tied. For $\mathrm{AUC}_{\mathrm{PR}}$ and PCA, the best $r$ is always smaller than the best $r$ for $|\mathrm{SRMSD}_p|$, so there is often a tradeoff between calibrated p-values versus classification performance. For LMM, there is no tradeoff, as $r = 0$ often has the best mean $\mathrm{AUC}_{\mathrm{PR}}$, and otherwise is not significantly different from the best $r$. Lastly, LMM with $r = 0$ always has significantly greater or statistically tied $\mathrm{AUC}_{\mathrm{PR}}$ than PCA with its best $r$.

## Evaluations in admixture simulations

Now we look more closely at results per dataset. The complete $\mathrm{SRMSD}_p$ and $\mathrm{AUC}_{\mathrm{PR}}$ distributions for the admixture simulations and FES traits are in *Figure 3*. RC traits gave qualitatively similar results (*Figure 3—figure supplement 1*).

In the large admixture simulation, the $\mathrm{SRMSD}_p$ of PCA is largest when $r = 0$ (no PCs) and decreases rapidly to near zero at $r = 3$, where it stays for up to $r = 90$ (*Figure 3A*). Thus, PCA has calibrated p-values for $r \geq 3$, smaller than the theoretical optimum for this simulation of $r = K - 1 = 9$. In contrast,

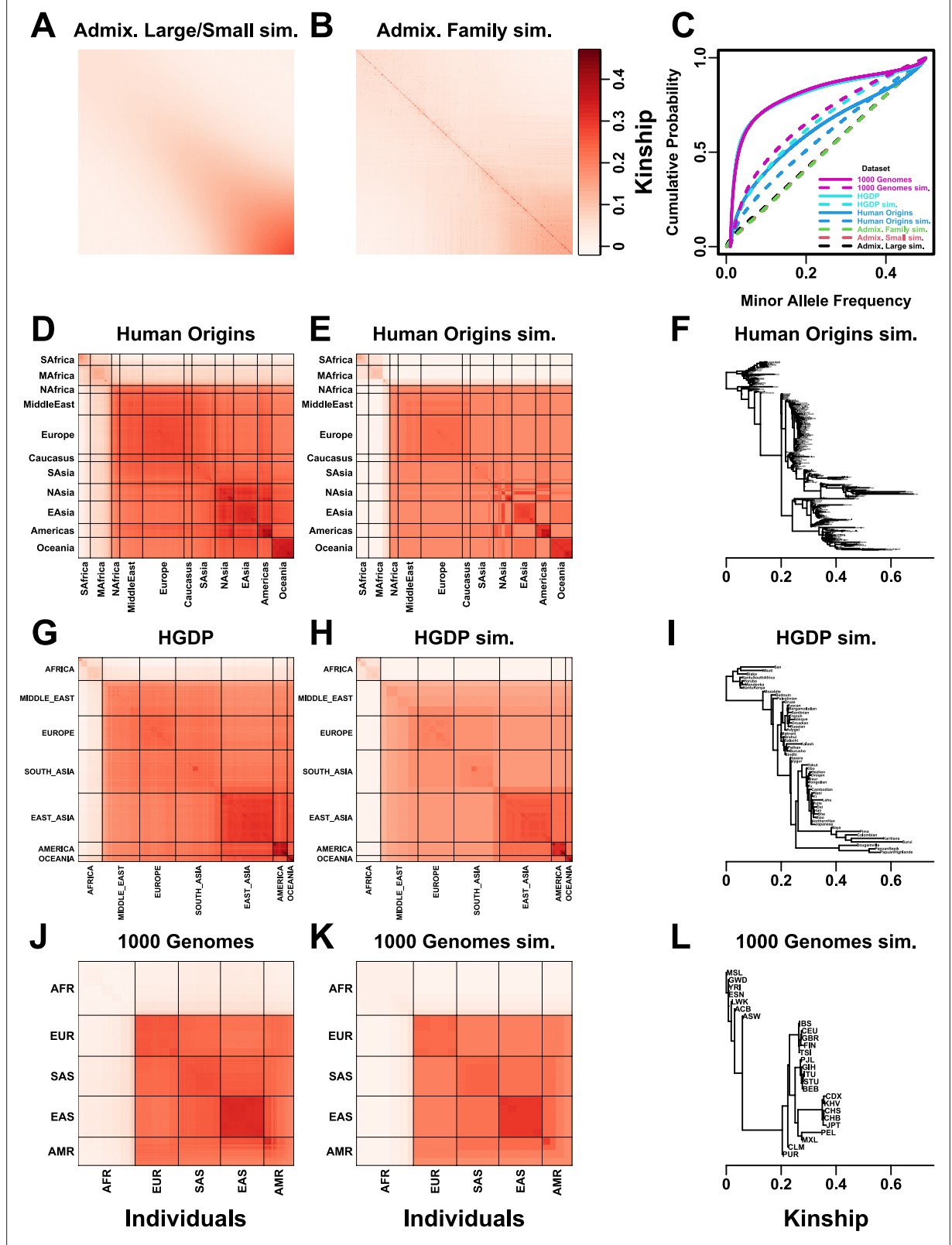

**Figure 1.** Population structures of simulated and real human genotype datasets. First two columns are population kinship matrices as heatmaps: individuals along x- and y-axis, kinship as color. Diagonal shows inbreeding values. (**A**) Admixture scenario for both Large and Small simulations. (**B**) Last generation of 20-generation admixed family, shows larger kinship values near diagonal corresponding to siblings, first cousins, etc. (**C**) Minor allele frequency (MAF) distributions. Real datasets and subpopulation tree simulations had $\mathrm{MAF} \geq 0.01$ filter. (**D**) Human Origins is an array dataset of a large

*Figure 1 continued on next page*

*Figure 1 continued*
diversity of global populations. (**G**) Human Genome Diversity Panel (HGDP) is a WGS dataset from global native populations. (**J**) 1000 Genomes Project is a WGS dataset of global cosmopolitan populations. (**F, I, L**) Trees between subpopulations fit to real data. (**E, H, K**). Simulations from trees fit to the real data recapitulate subpopulation structure.

the $\text{SRMSD}_p$ for LMM starts near zero for $r = 0$, but becomes negative as $r$ increases (p-values are conservative). The $\text{AUC}_{\text{PR}}$ distribution of PCA is similarly worst at $r = 0$, increases rapidly and peaks at $r = 3$, then decreases slowly for $r > 3$, while the $\text{AUC}_{\text{PR}}$ distribution for LMM starts near its maximum at $r = 0$ and decreases with $r$. Although the $\text{AUC}_{\text{PR}}$ distributions for LMM and PCA overlap considerably at each $r$, LMM with $r = 0$ has significantly greater $\text{AUC}_{\text{PR}}$ values than PCA with $r = 3$ (*Table 3*). However, qualitatively PCA performs nearly as well as LMM in this simulation.

The observed robustness to large $r$ led us to consider smaller sample sizes. A model with large numbers of parameters $r$ should overfit more as $r$ approaches the sample size $n$. Rather than increase $r$ beyond 90, we reduce individuals to $n = 100$, which is small for typical association studies but may occur in studies of rare diseases, pilot studies, or other constraints. To compensate for the loss of power

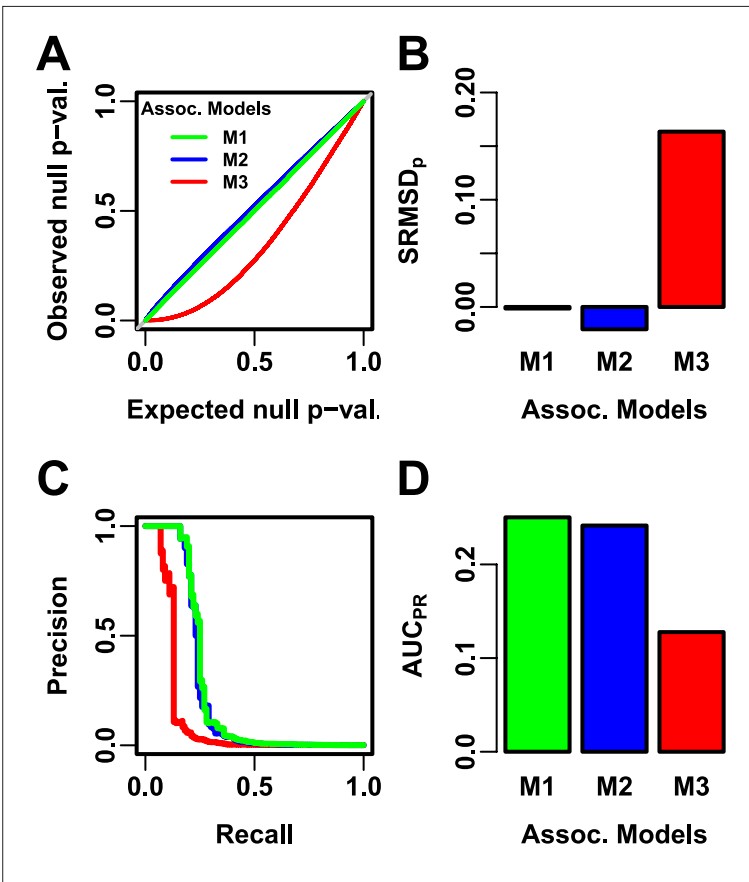

**Figure 2.** Illustration of evaluation measures. Three archetypal models illustrate our complementary measures: M1 is ideal, M2 overfits slightly, M3 is naive. (**A**) QQ plot of p-values of "null" (non-causal) loci. M1 has desired uniform p-values, M2/M3 are miscalibrated. (**B**)$\text{SRMSD}_p$ (p-value Signed Root Mean Square Deviation) measures signed distance between observed and expected null p-values (closer to zero is better). (**C**) Precision and Recall (PR) measure causal locus classification performance (higher is better). (**D**) $\text{AUC}_{\text{PR}}$ (Area Under the PR Curve) reflects power (higher is better).

The online version of this article includes the following figure supplement(s) for figure 2:

**Figure supplement 1.** Comparison between $\text{SRMSD}_p$ and inflation factor.

**Figure supplement 2.** Comparison between $\text{SRMSD}_p$ and type I error rate.

**Figure supplement 3.** Comparison between $\text{AUC}_{\text{PR}}$ and calibrated power.

**Table 3.** Overview of PCA and LMM evaluations for high heritability simulations.

| | | | LMM $r = 0$ vs best $r$ | | | PCA vs LMM $r = 0$ | | | |
| Dataset | Metric | Trait* | Cal.† | Best $r$‡ | P-value § | Best $r$‡ | Cal.† | P-value § | Best model ¶ |
|---|---|---|---|---|---|---|---|---|---|
| Admix. Large sim. | \|SRMSD$_p$\| | FES | True | 0 | 1 | 12 | True | 0.036 | Tie |
| Admix. Small sim. | \|SRMSD$_p$\| | FES | True | 0 | 1 | 4 | True | 0.055 | Tie |
| Admix. Family sim. | \|SRMSD$_p$\| | FES | True | 0 | 1 | 90 | False | 3.9e-10* | LMM |
| Human Origins | \|SRMSD$_p$\| | FES | True | 0 | 1 | 89 | False | 3.9e-10* | LMM |
| HGDP | \|SRMSD$_p$\| | FES | True | 0 | 1 | 87 | True | 4.4e-10* | LMM |
| 1000 Genomes | \|SRMSD$_p$\| | FES | True | 0 | 1 | 90 | False | 3.9e-10* | LMM |
| Human Origins sim. | \|SRMSD$_p$\| | FES | True | 0 | 1 | 88 | True | 0.017 | Tie |
| HGDP sim. | \|SRMSD$_p$\| | FES | True | 0 | 1 | 47 | True | 0.046 | Tie |
| 1000 Genomes sim. | \|SRMSD$_p$\| | FES | True | 0 | 1 | 78 | True | 9.6e-10* | LMM |
| Admix. Large sim. | \|SRMSD$_p$\| | RC | True | 0 | 1 | 26 | True | 0.11 | Tie |
| Admix. Small sim. | \|SRMSD$_p$\| | RC | True | 0 | 1 | 4 | True | 0.00097 | Tie |
| Admix. Family sim. | \|SRMSD$_p$\| | RC | True | 0 | 1 | 90 | False | 3.9e-10* | LMM |
| Human Origins | \|SRMSD$_p$\| | RC | True | 0 | 1 | 90 | True | 0.00065 | Tie |
| HGDP | \|SRMSD$_p$\| | RC | True | 0 | 1 | 37 | True | 1.5e-05* | LMM |
| 1000 Genomes | \|SRMSD$_p$\| | RC | True | 0 | 1 | 76 | True | 3.9e-10* | LMM |
| Human Origins sim. | \|SRMSD$_p$\| | RC | True | 0 | 1 | 85 | True | 0.14 | Tie |
| HGDP sim. | \|SRMSD$_p$\| | RC | True | 0 | 1 | 44 | True | 8.8e-07* | LMM |
| 1000 Genomes sim. | \|SRMSD$_p$\| | RC | True | 0 | 1 | 90 | True | 3.9e-10* | LMM |
| Admix. Large sim. | AUC$_{PR}$ | FES | | 0 | 1 | 3 | | 5.9e-06* | LMM |
| Admix. Small sim. | AUC$_{PR}$ | FES | | 0 | 1 | 2 | | 0.025 | Tie |
| Admix. Family sim. | AUC$_{PR}$ | FES | | 1 | 0.35 | 22 | | 3.9e-10* | LMM |
| Human Origins | AUC$_{PR}$ | FES | | 0 | 1 | 34 | | 3.9e-10* | LMM |
| HGDP | AUC$_{PR}$ | FES | | 1 | 0.33 | 16 | | 4.4e-10* | LMM |
| 1000 Genomes | AUC$_{PR}$ | FES | | 1 | 0.11 | 8 | | 3.9e-10* | LMM |
| Human Origins sim. | AUC$_{PR}$ | FES | | 0 | 1 | 36 | | 3.9e-10* | LMM |
| HGDP sim. | AUC$_{PR}$ | FES | | 0 | 1 | 17 | | 1.7e-05* | LMM |
| 1000 Genomes sim. | AUC$_{PR}$ | FES | | 0 | 1 | 10 | | 5e-10* | LMM |
| Admix. Large sim. | AUC$_{PR}$ | RC | | 0 | 1 | 3 | | 1.4e-05* | LMM |
| Admix. Small sim. | AUC$_{PR}$ | RC | | 0 | 1 | 1 | | 0.095 | Tie |
| Admix. Family sim. | AUC$_{PR}$ | RC | | 0 | 1 | 34 | | 3.9e-10* | LMM |
| Human Origins | AUC$_{PR}$ | RC | | 3 | 0.4 | 36 | | 9.6e-10* | LMM |
| HGDP | AUC$_{PR}$ | RC | | 4 | 0.21 | 16 | | 0.013 | Tie |
| 1000 Genomes | AUC$_{PR}$ | RC | | 5 | 0.004 | 9 | | 0.00043 | Tie |
| Human Origins sim. | AUC$_{PR}$ | RC | | 0 | 1 | 37 | | 4.1e-10* | LMM |

*Table 3 continued on next page*

*Table 3 continued*

| | | | LMM $r = 0$ vs best $r$ | | | PCA vs LMM $r = 0$ | | | |
|---|---|---|---|---|---|---|---|---|---|
| Dataset | Metric | Trait* | Cal.† | Best $r$‡ | P-value § | Best $r$‡ | Cal.† | P-value § | Best model ¶ |
| HGDP sim. | $\mathrm{AUC_{PR}}$ | RC | | 3 | 0.087 | 17 | | 0.0014 | Tie |
| 1000 Genomes sim. | $\mathrm{AUC_{PR}}$ | RC | | 3 | 0.37 | 10 | | 8.5e-10* | LMM |

*FES: Fixed Effect Sizes, RC: Random Coefficients.

†Calibrated: whether mean $|\mathrm{SRMSD}_p| < 0.01$ over 50 replicates.

‡Value of $r$ (number of PCs) with minimum mean $|\mathrm{SRMSD}_p|$ or maximum mean $\mathrm{AUC_{PR}}$.

§Wilcoxon paired 1-tailed test of distributions ($|\mathrm{SRMSD}_p|$ or $\mathrm{AUC_{PR}}$) between models in header. Asterisk marks significant value using Bonferroni threshold ($p < \alpha/n_\mathrm{tests}$ with $\alpha = 0.01$ and $n_\mathrm{tests} = 72$ is the number of tests in this table).

¶Tie if no significant difference using Bonferroni threshold.

due to reducing $n$, we also reduce the number of causal loci (see Trait Simulation), which increases per-locus effect sizes. We found a large decrease in performance for both models as $r$ increases, and best performance for $r = 1$ for PCA and $r = 0$ for LMM (*Figure 3B*). Remarkably, LMM attains much larger negative $\mathrm{SRMSD}_p$ values than in our other evaluations. LMM with $r = 0$ is significantly better than PCA ($r = 1$ to 4) in both measures (*Table 3*), but qualitatively the difference is negligible.

The family simulation adds a 20-generation random family to our large admixture simulation. Only the last generation is studied for association, which contains numerous siblings, first cousins, etc., with the initial admixture structure preserved by geographically biased mating. Our evaluation reveals a sizable gap in both measures between LMM and PCA across all $r$ (*Figure 3C*). LMM again performs best with $r = 0$ and achieves mean $|\mathrm{SRMSD}_p| < 0.01$. However, PCA does not achieve mean $|\mathrm{SRMSD}_p| < 0.01$ at any $r$, and its best mean $\mathrm{AUC_{PR}}$ is considerably worse than that of LMM. Thus, LMM is conclusively superior to PCA, and the only calibrated model, when there is family structure.

## Evaluations in real human genotype datasets

Next, we repeat our evaluations with real human genotype data, which differs from our simulations in allele frequency distributions and more complex population structures with greater $F_\mathrm{ST}$, numerous correlated subpopulations, and potential cryptic family relatedness.

Human Origins has the greatest number and diversity of subpopulations. The $\mathrm{SRMSD}_p$ and $\mathrm{AUC_{PR}}$ distributions in this dataset and FES traits (*Figure 4A*) most resemble those from the family simulation (*Figure 3C*). In particular, while LMM with $r = 0$ performed optimally (both measures) and satisfies mean $|\mathrm{SRMSD}_p| < 0.01$, PCA maintained $\mathrm{SRMSD}_p > 0.01$ for all $r$ and its $\mathrm{AUC_{PR}}$ were all considerably smaller than the best $\mathrm{AUC_{PR}}$ of LMM.

HGDP has the fewest individuals among real datasets, but compared to Human Origins contains more loci and low-frequency variants. Performance (*Figure 4B*) again most resembled the family simulations. In particular, LMM with $r = 0$ achieves mean $|\mathrm{SRMSD}_p| < 0.01$ (p-values are calibrated), while PCA does not, and there is a sizable $\mathrm{AUC_{PR}}$ gap between LMM and PCA. Maximum $\mathrm{AUC_{PR}}$ values were lowest in HGDP compared to the two other real datasets.

1000 Genomes has the fewest subpopulations but largest number of individuals per subpopulation. Thus, although this dataset has the simplest subpopulation structure among the real datasets, we find $\mathrm{SRMSD}_p$ and $\mathrm{AUC_{PR}}$ distributions (*Figure 4C*) that again most resemble our earlier family simulation, with mean $|\mathrm{SRMSD}_p| < 0.01$ for LMM only and large $\mathrm{AUC_{PR}}$ gaps between LMM and PCA.

Our results are qualitatively different for RC traits, which had smaller $\mathrm{AUC_{PR}}$ gaps between LMM and PCA (*Figure 4—figure supplement 1*). Maximum $\mathrm{AUC_{PR}}$ were smaller in RC compared to FES in Human Origins and 1000 Genomes, suggesting lower power for RC traits across association models. Nevertheless, LMM with $r = 0$ was significantly better than PCA for all measures in the real datasets and RC traits (*Table 3*).

## Evaluations in subpopulation tree simulations fit to human data

To better understand which features of the real datasets lead to the large differences in performance between LMM and PCA, we carried out subpopulation tree simulations. Human subpopulations are related roughly by trees, which induce the strongest correlations, so we fit trees to each real dataset

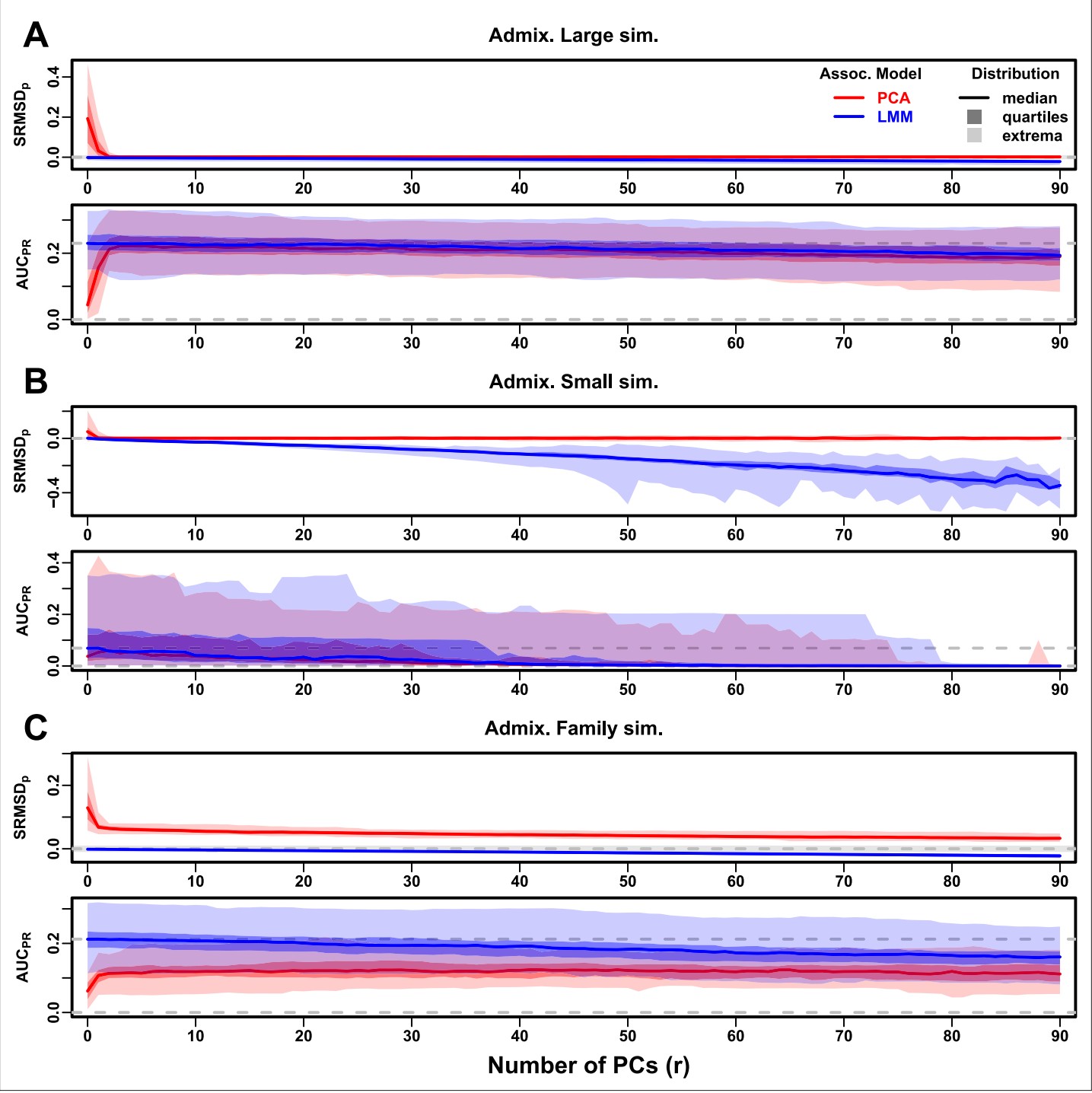

**Figure 3.** Evaluations in admixture simulations with FES traits, high heritability. PCA and LMM models have varying number of PCs ($r \in \{0, ..., 90\}$ on x-axis), with the distributions (y-axis) of $\mathrm{SRMSD}_p$ (top subpanel) and $\mathrm{AUC_{PR}}$ (bottom subpanel) for 50 replicates. Best performance is zero $\mathrm{SRMSD}_p$ and large $\mathrm{AUC_{PR}}$. Zero and maximum median $\mathrm{AUC_{PR}}$ values are marked with horizontal gray dashed lines, and $|\mathrm{SRMSD}_p| < 0.01$ is marked with a light gray area. LMM performs best with $r = 0$, PCA with various $r$. (**A**) Large simulation ($n = 1,000$ individuals). (**B**) Small simulation ($n = 100$) shows overfitting for large $r$. (**C**) Family simulation ($n = 1,000$) has admixed founders and large numbers of close relatives from a realistic random 20-generation pedigree. PCA performs poorly compared to LMM: $\mathrm{SRMSD}_p > 0$ for all $r$ and large $\mathrm{AUC_{PR}}$ gap.

The online version of this article includes the following figure supplement(s) for figure 3:

**Figure supplement 1.** Evaluations in admixture simulations with RC traits, high heritability.

**Figure supplement 2.** Evaluations in admixture simulations with FES traits, low heritability.

*Figure 3 continued on next page*

and tested if data simulated from these complex tree structures could recapitulate our previous results (*Figure 1*). These tree simulations also feature non-uniform ancestral allele frequency distributions, which recapitulated some of the skew for smaller minor allele frequencies of the real datasets (*Figure 1C*). The $SRMSD_p$ and $AUC_{PR}$ distributions for these tree simulations (*Figure 5*) resembled our admixture simulation more than either the family simulation (*Figure 3*) or real data results (*Figure 4*). Both LMM with $r = 0$ and PCA (various $r$) achieve mean $|SRMSD_p| < 0.01$ (*Table 3*). The $AUC_{PR}$ distributions of both LMM and PCA track closely as $r$ is varied, although there is a small gap resulting in LMM ($r = 0$) besting PCA in all three simulations. The results are qualitatively similar for RC traits (*Figure 5—figure supplement 1*, *Table 3*). Overall, these subpopulation tree simulations do not recapitulate the large LMM advantage over PCA observed on the real data.

## Numerous distant relatives explain poor PCA performance in real data

In principle, PCA performance should be determined by the dimension of relatedness, or kinship matrix rank, since PCA is a low-dimensional model whereas LMM can model high-dimensional relatedness without overfitting. We used the Tracy-Widom test (*Patterson et al., 2006*) with $p < 0.01$ to estimate kinship matrix rank as the number of significant PCs (*Figure 6—figure supplement 1A*). The true rank of our simulations is slightly underestimated (*Table 2*), but we confirm that the family simulation has the greatest rank, and real datasets have greater estimates than their respective subpopulation tree simulations, which confirms our hypothesis to some extent. However, estimated ranks do not separate real datasets from tree simulations, as required to predict the observed PCA performance. Moreover, the HGDP and 1000 Genomes rank estimates are 45 and 61, respectively, yet PCA performed poorly for all $r \leq 90$ numbers of PCs (*Figure 4*). The top eigenvalue explained a proportion of variance proportional to $F_{ST}$ (*Table 2*), but the rest of the top 10 eigenvalues show no clear differences between datasets, except the small simulation had larger variances explained per eigenvalue (expected since it has fewer eigenvalues; *Figure 6—figure supplement 1*). Comparing cumulative variance explained versus rank fraction across all eigenvalues, all datasets increase from their starting point almost linearly until they reach 1, except the family simulation has much greater variance explained by mid-rank eigenvalues (*Figure 6—figure supplement 1*). We also calculated the number of PCs that are significantly associated with the trait, and observed similar results, namely that while the family simulation has more significant PCs than the non-family admixture simulations, the real datasets and their tree simulated counterparts have similar numbers of significant PCs (*Figure 6—figure supplement 2*). Overall, there is no separation between real datasets (where PCA performed poorly) and subpopulation tree simulations (where PCA performed relatively well) in terms of their eigenvalues or kinship matrix rank estimates.

Local kinship, which is recent relatedness due to family structure excluding population structure, is the presumed cause of the LMM to PCA performance gap observed in real datasets but not their subpopulation tree simulation counterparts. Instead of inferring local kinship through increased kinship matrix rank, as attempted in the last paragraph, now we measure it directly using the KING-robust estimator (*Manichaikul et al., 2010*). We observe more large local kinship in the real datasets and the family simulation compared to the other simulations (*Figure 6*). However, for real data this distribution depends on the subpopulation structure, since locally related pairs are most likely in the same subpopulation. Therefore, the only comparable curve to each real dataset is their corresponding subpopulation tree simulation, which matches subpopulation structure. In all real datasets, we identified highly related individual pairs with kinship above the 4th degree relative threshold of 0.022 (*Manichaikul et al., 2010*; *Conomos et al., 2016b*). However, these highly related pairs are vastly outnumbered by more distant pairs with evident non-zero local kinship as compared to the extreme tree simulation values.

To try to improve PCA performance, we followed the standard practice of removing 4th degree relatives, which reduced sample sizes between 5% and 10% (*Table 4*). Only $r = 0$ for LMM and $r = 20$

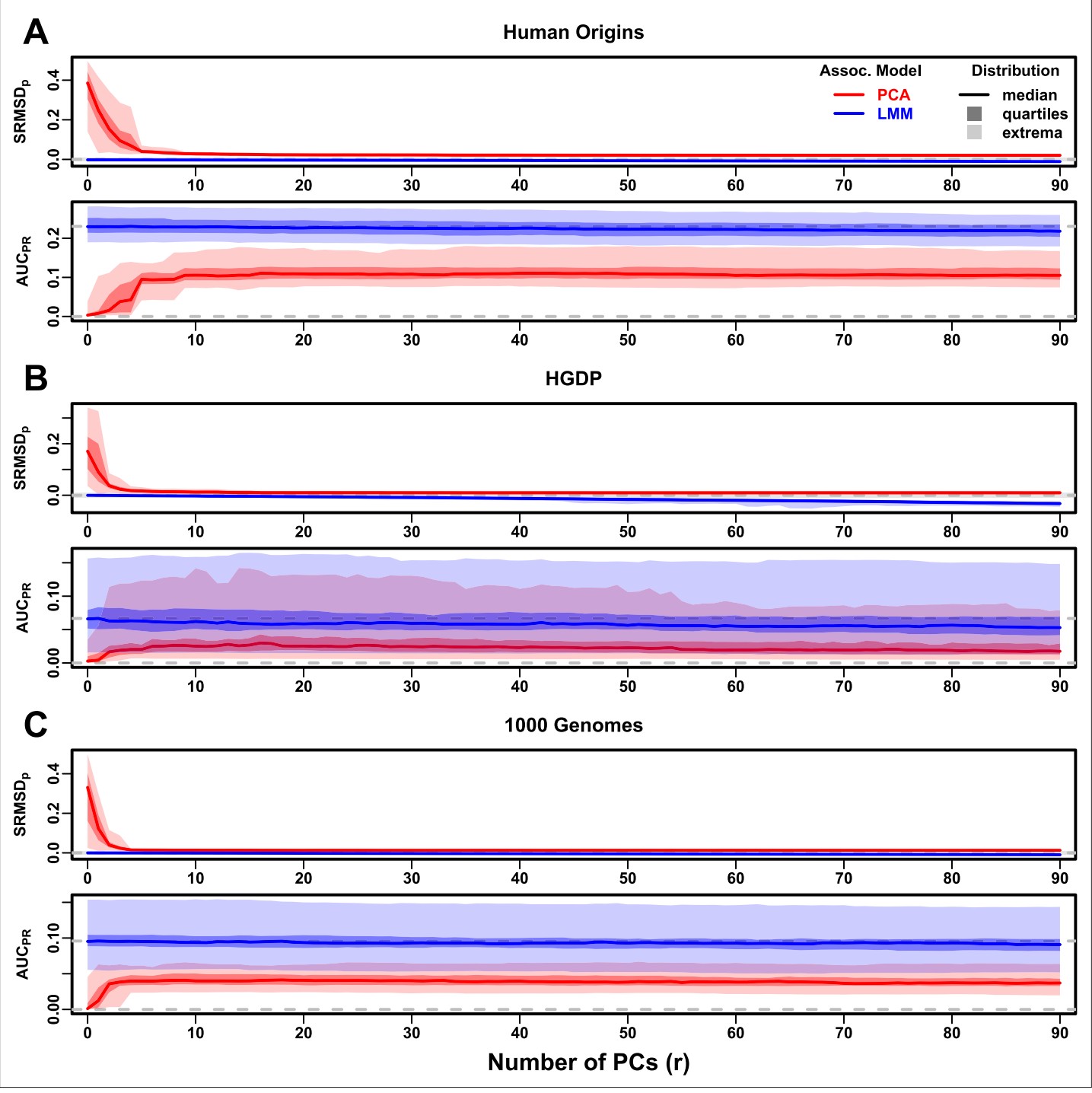

**Figure 4.** Evaluations in real human genotype datasets with FES traits, high heritability. Same setup as *Figure 3*, see that for details. These datasets strongly favor LMM with no PCs over PCA, with distributions that most resemble the family simulation. (**A**) Human Origins. (**B**) Human Genome Diversity Panel (HGDP). (**C**) 1000 Genomes Project.

The online version of this article includes the following figure supplement(s) for figure 4:

**Figure supplement 1.** Evaluations in real human genotype datasets with RC traits, high heritability.

**Figure supplement 2.** Evaluations in real human genotype datasets with FES traits, low heritability.

**Figure supplement 3.** Evaluations in real human genotype datasets with RC traits, low heritability.

**Figure supplement 4.** Evaluations in real human genotype datasets with FES traits, environment.

**Figure supplement 5.** Evaluations in real human genotype datasets with RC traits, environment.

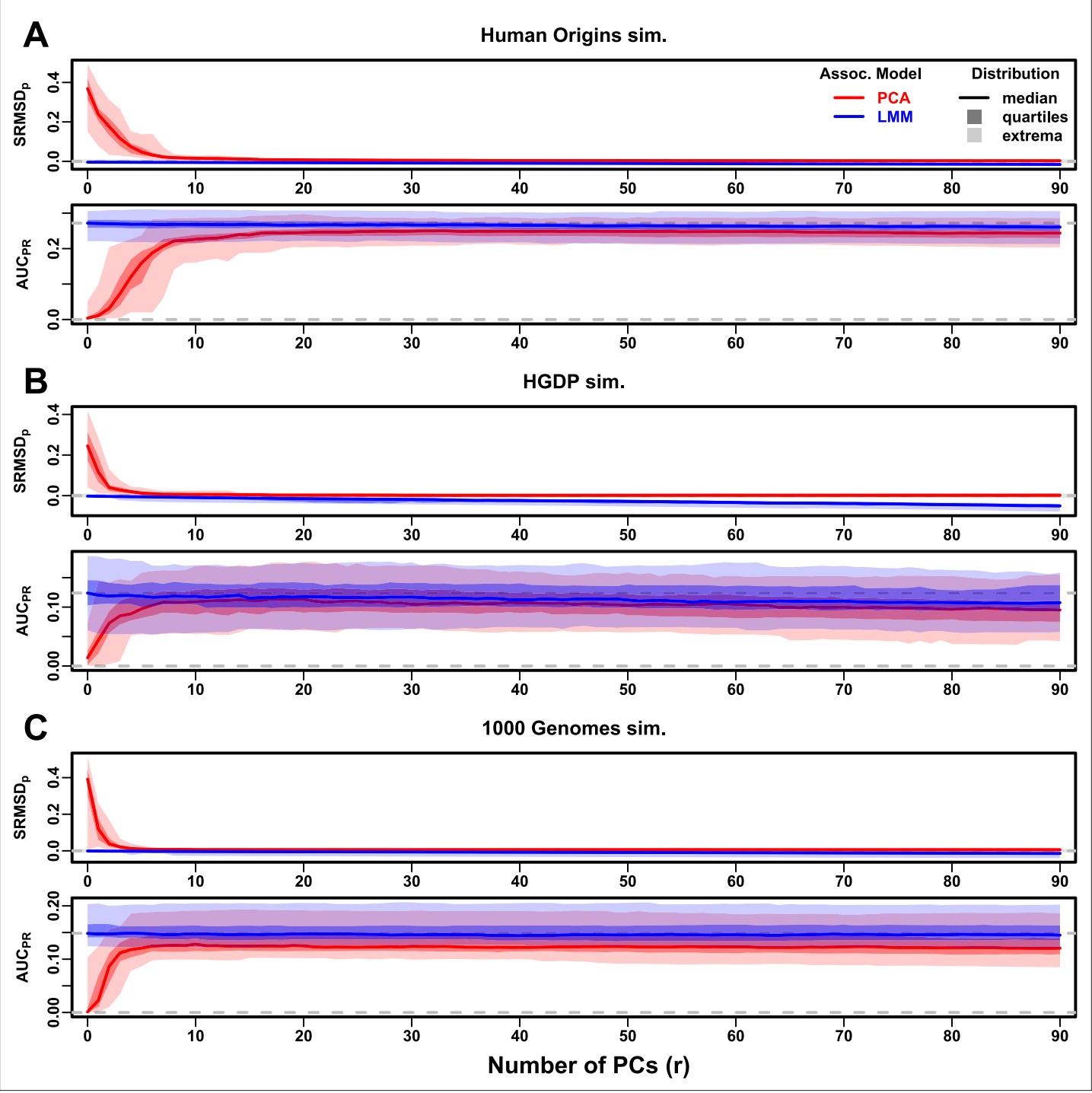

**Figure 5.** Evaluations in subpopulation tree simulations fit to human data with FES traits, high heritability. Same setup as *Figure 3*, see that for details. These tree simulations, which exclude family structure by design, do not explain the large gaps in LMM-PCA performance observed in the real data. (**A**) Human Origins tree simulation. (**B**) Human Genome Diversity Panel (HGDP) tree simulation. (**C**) 1000 Genomes Project tree simulation.

The online version of this article includes the following figure supplement(s) for figure 5:

**Figure supplement 1.** Evaluations in subpopulation tree simulations fit to human data with RC traits, high heritability.

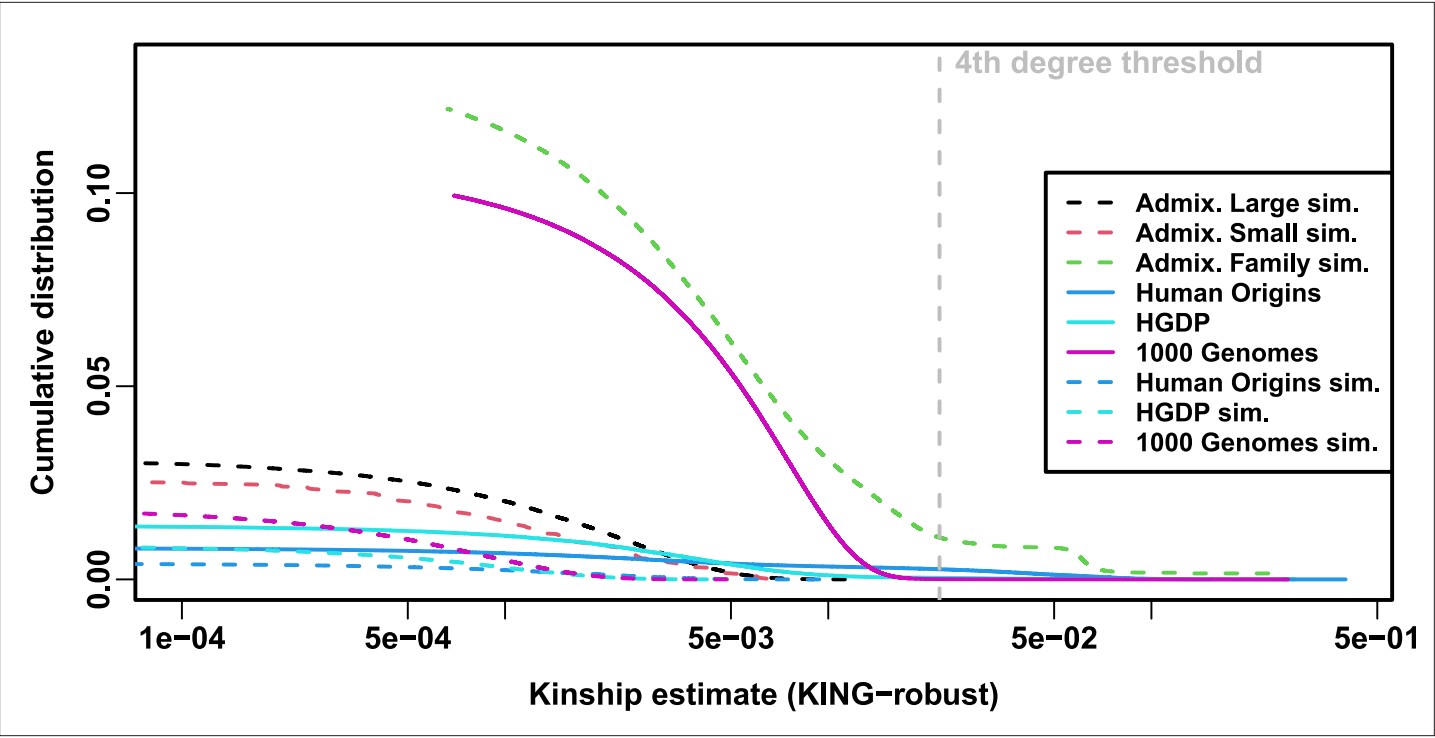

**Figure 6.** Local kinship distributions. Curves are complementary cumulative distribution of lower triangular kinship matrix (self kinship excluded) from KING-robust estimator. Note log x-axis; negative estimates are counted but not shown. Most values are below 4th degree relative threshold. Each real dataset has a greater cumulative than its subpopulation tree simulations.

The online version of this article includes the following figure supplement(s) for figure 6:

**Figure supplement 1.** Estimated relatedness dimensions of datasets.

**Figure supplement 2.** Number of PCs significantly associated with traits.

for PCA were tested, as these performed well in our earlier evaluation, and only FES traits were tested because they previously displayed the large PCA-LMM performance gap. LMM significantly outperforms PCA in all these cases (Wilcoxon paired 1-tailed $p < 0.01$; *Figure 7*). Notably, PCA still had miscalibrated p-values two of the three real datasets ($|SRMSD_p| > 0.01$), the only marginally calibrated case being HGDP which is also the smallest of these datasets. Otherwise, $AUC_{PR}$ and $SRMSD_p$ ranges were similar here as in our earlier evaluation. Therefore, the removal of the small number of highly related individual pairs had a negligible effect in PCA performance, so the larger number of more distantly related pairs explain the poor PCA performance in the real datasets.

### Low heritability and environment simulations

Our main evaluations were repeated with traits simulated under a lower heritability value of $h^2 = 0.3$. We reduced the number of causal loci in response to this change in heritability, to result in equal average effect size per locus compared to the previous high heritability evaluations (see Trait Simulation). Despite that, these low heritability evaluations measured lower $AUC_{PR}$ values than their high heritability counterparts (*Figure 3—figure supplement 2*, *Figure 3—figure supplement 3*, *Figure 4—figure supplement 2*, *Figure 4—figure supplement 3*, *Figure 7—figure supplement 1*).

**Table 4.** Dataset sizes after 4th degree relative filter.

| Dataset | Loci ($m$) | Ind. ($n$) | Ind. removed (%) |
|---|---|---|---|
| Human Origins | 189 722 | 2636 | 9.8 |
| HGDP | 758 009 | 847 | 8.8 |
| 1000 Genomes | 1 097 415 | 2390 | 4.6 |

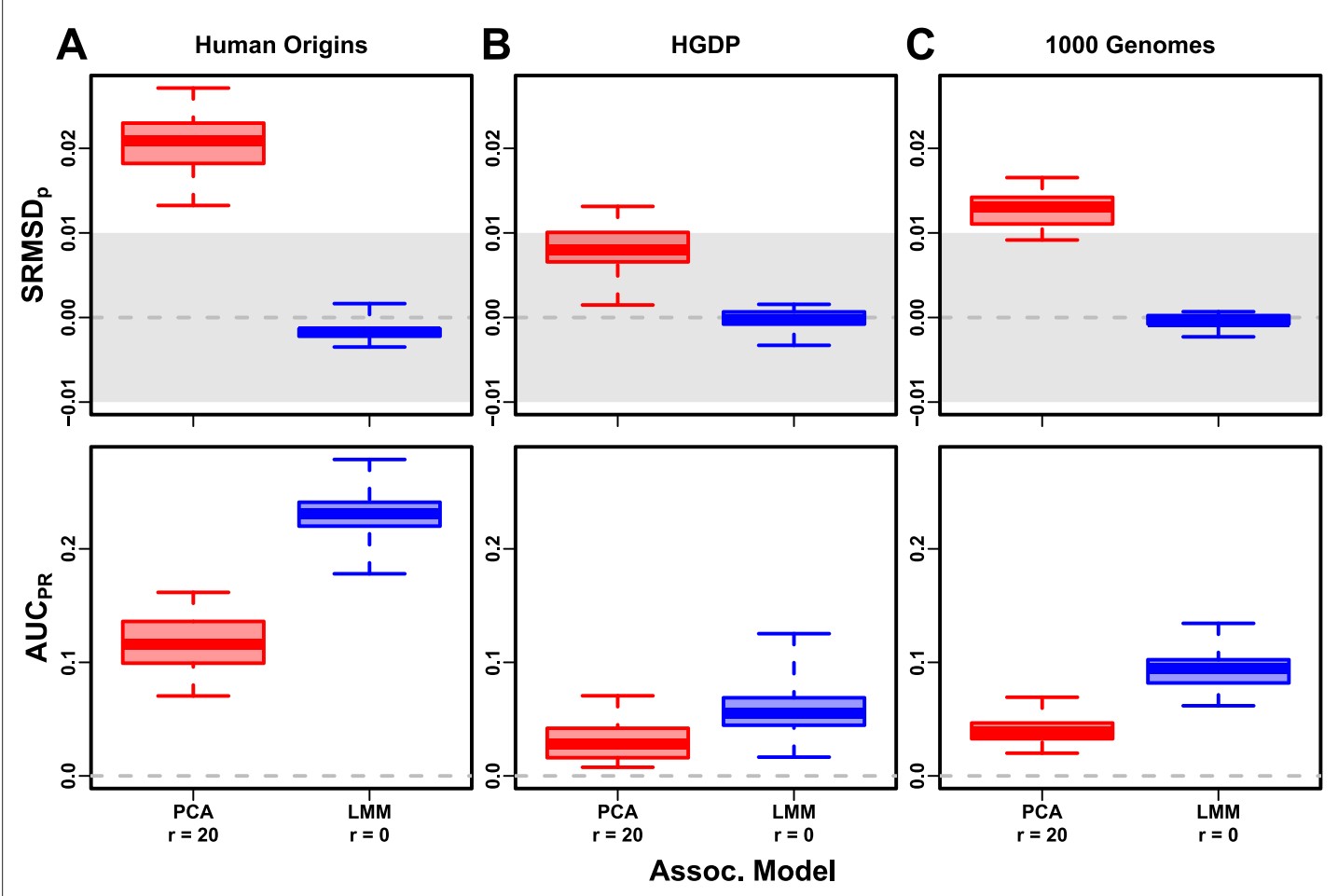

**Figure 7.** Evaluation in real datasets excluding 4th degree relatives, FES traits, high heritability. Each dataset is a column, rows are measures. Boxplot whiskers are extrema over 50 replicates. First row has |SRMSD$_p$| < 0.01 band marked as gray area.

The online version of this article includes the following figure supplement(s) for figure 7:

**Figure supplement 1.** Evaluation in real datasets excluding 4th degree relatives, FES traits, low heritability.

The gap between LMM and PCA was reduced in these evaluations, but the main conclusion of the high heritability evaluation holds for low heritability as well, namely that LMM with $r = 0$ significantly outperforms or ties LMM with $r > 0$ and PCA in all cases (*Table 5*).

Lastly, we simulated traits with both low heritability and large environment effects determined by geography and subpopulation labels, so they are strongly correlated to the low-dimensional population structure. For that reason, PCs may be expected to perform better in this setting (in either PCA or LMM). However, we find that both PCA and LMM (even without PCs) increase their AUC$_{PR}$ values compared to the low-heritability evaluations (*Figure 8—figure supplement 1*; *Figure 8* also shows representative numbers of PCs, which performed optimally or nearly so in individual simulations shown in *Figure 3—figure supplement 4*, *Figure 3—figure supplement 5*, *Figure 4—figure supplement 4*, *Figure 4—figure supplement 5*). p-Value calibration is comparable with or without environment effects, for LMM for all $r$ and for PCA once $r$ is large enough (*Figure 8—figure supplement 1*). These simulations are the only where we occasionally observed for both metrics a significant, though small, advantage of LMM with PCs versus LMM without PCs (*Table 6*). Additionally, on RC traits only, PCA significantly outperforms LMM in the three real human datasets (*Table 6*), the only cases in all of our evaluations where this is observed. For comparison, we also evaluate an 'oracle' LMM without PCs but with the finest group labels, the same used to simulate environment, as fixed categorical covariates ('LMM lab.'), and see much larger AUC$_{PR}$ values than either LMM with PCs or PCA (*Figure 8*, *Figure 3—figure supplement 4*, *Figure 3—figure supplement 5*, *Figure 4—figure*

**Table 5.** Overview of PCA and LMM evaluations for low heritability simulations.

| Dataset | Metric | Trait* | LMM $r = 0$ vs best $r$ | | | PCA vs LMM $r = 0$ | | | Best model [¶] |
|---|---|---|---|---|---|---|---|---|---|
| | | | Cal.[†] | Best $r$[‡] | p-value [§] | Best $r$[‡] | Cal.[†] | p-value [§] | |
| Admix. Large sim. | \|SRMSD$_p$\| | FES | True | 0 | 1 | 62 | True | 0.00012* | LMM |
| Admix. Small sim. | \|SRMSD$_p$\| | FES | True | 0 | 1 | 3 | True | 0.27 | Tie |
| Admix. Family sim. | \|SRMSD$_p$\| | FES | True | 0 | 1 | 90 | False | 3.9e-10* | LMM |
| Human Origins | \|SRMSD$_p$\| | FES | True | 0 | 1 | 81 | True | 3.9e-10* | LMM |
| HGDP | \|SRMSD$_p$\| | FES | True | 0 | 1 | 37 | True | 6.2e-09* | LMM |
| 1000 Genomes | \|SRMSD$_p$\| | FES | True | 0 | 1 | 84 | True | 3.9e-10* | LMM |
| Admix. Large sim. | \|SRMSD$_p$\| | RC | True | 0 | 1 | 35 | True | 0.00094 | Tie |
| Admix. Small sim. | \|SRMSD$_p$\| | RC | True | 0 | 1 | 3 | True | 0.087 | Tie |
| Admix. Family sim. | \|SRMSD$_p$\| | RC | True | 0 | 1 | 90 | False | 4.1e-10* | LMM |
| Human Origins | \|SRMSD$_p$\| | RC | True | 0 | 1 | 75 | True | 0.00016* | LMM |
| HGDP | \|SRMSD$_p$\| | RC | True | 0 | 1 | 23 | True | 1.7e-05* | LMM |
| 1000 Genomes | \|SRMSD$_p$\| | RC | True | 0 | 1 | 41 | True | 6.7e-10* | LMM |
| Admix. Large sim. | AUC$_{PR}$ | FES | | 0 | 1 | 3 | | 0.11 | Tie |
| Admix. Small sim. | AUC$_{PR}$ | FES | | 0 | 1 | 0 | | 0.58 | Tie |
| Admix. Family sim. | AUC$_{PR}$ | FES | | 0 | 1 | 7 | | 2.2e-06* | LMM |
| Human Origins | AUC$_{PR}$ | FES | | 0 | 1 | 16 | | 8e-10* | LMM |
| HGDP | AUC$_{PR}$ | FES | | 11 | 0.68 | 6 | | 0.0043 | Tie |
| 1000 Genomes | AUC$_{PR}$ | FES | | 6 | 0.34 | 4 | | 2.3e-07* | LMM |
| Admix. Large sim. | AUC$_{PR}$ | RC | | 0 | 1 | 3 | | 0.14 | Tie |
| Admix. Small sim. | AUC$_{PR}$ | RC | | 0 | 1 | 0 | | 0.1 | Tie |
| Admix. Family sim. | AUC$_{PR}$ | RC | | 0 | 1 | 5 | | 1.9e-06* | LMM |
| Human Origins | AUC$_{PR}$ | RC | | 4 | 0.16 | 12 | | 0.003 | Tie |
| HGDP | AUC$_{PR}$ | RC | | 2 | 0.14 | 5 | | 0.14 | Tie |
| 1000 Genomes | AUC$_{PR}$ | RC | | 0 | 1 | 4 | | 0.078 | Tie |

*FES: Fixed Effect Sizes, RC: Random Coefficients.

[†]Calibrated: whether mean \|SRMSD$_p$\| < 0.01 over 50 replicates.

[‡]Value of $r$ (number of PCs) with minimum mean \|SRMSD$_p$\| or maximum mean AUC$_{PR}$.

[§]Wilcoxon paired 1-tailed test of distributions (\|SRMSD$_p$\| or AUC$_{PR}$) between models in header. Asterisk marks significant value using Bonferroni threshold ($p < \alpha/n_{tests}$ with $\alpha = 0.01$ and $n_{tests} = 48$ is the number of tests in this table).

[¶]Tie if no significant difference using Bonferroni threshold.

*supplement 4*, *Figure 4—figure supplement 5*, *Table 6*). However, LMM with labels is often more poorly calibrated than LMM or PCA without labels, which may be since these numerous labels are inappropriately modeled as fixed rather than random effects. Overall, we find that association studies with correlated environment and genetic effects remain a challenge for PCA and LMM, that addition of PCs to an LMM improves performance only marginally, and that if the environment effect is driven by geography or ethnicity then use of those labels greatly improves performance compared to using PCs.

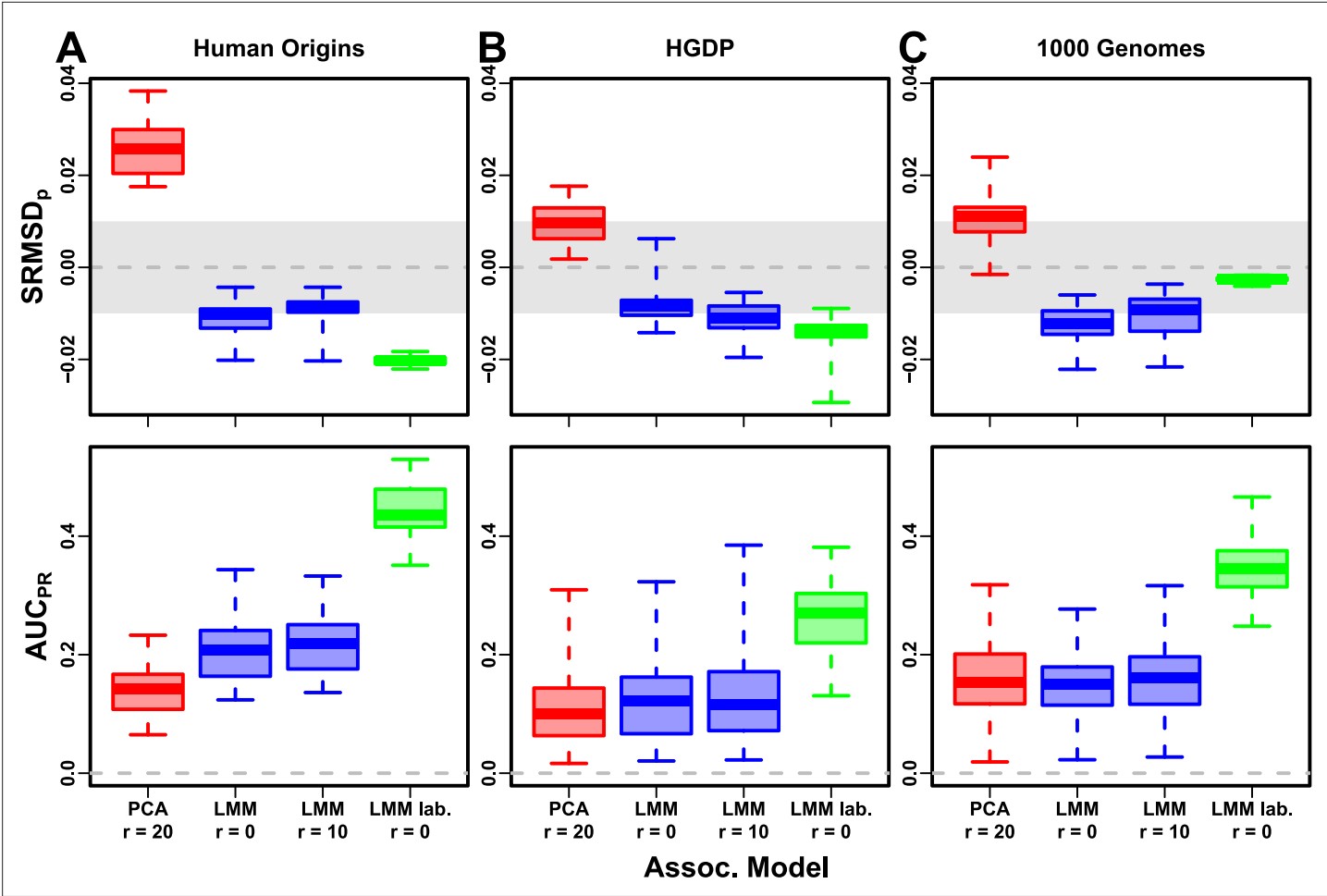

**Figure 8.** Evaluation in real datasets excluding 4th degree relatives, FES traits, environment. Traits simulated with environment effects, otherwise the same as *Figure 7*. 'LMM lab.' includes as fixed effects true groups from which environment was simulated.

The online version of this article includes the following figure supplement(s) for figure 8:

**Figure supplement 1.** Comparison of performance in low heritability vs environment simulations.

## Discussion

Our evaluations conclusively determined that LMM without PCs performs better than PCA (for any number of PCs) across all scenarios without environment effects, including all real and simulated genotypes and two trait simulation models. Although the addition of a few PCs to LMM does not greatly hurt its performance (except for small sample sizes), they generally did not improve it either (*Table 3*, *Table 5*), which agrees with previous observations (*Liu et al., 2011*; *Janss et al., 2012*) but contradicts others (*Zhao et al., 2007*; *Price et al., 2010*). Our findings make sense since PCs are the eigenvectors of the same kinship matrix that parameterized random effects, so including both is redundant.

The presence of environment effects that are correlated to relatedness presents the only scenario where occasionally PCA and LMM with PCs outperform LMM without PCs (*Table 6*). It is commonly believed that PCs model such environment effects well (*Novembre et al., 2008*; *Zhang and Pan, 2015*; *Lin et al., 2021*). However, we observe that LMM without PCs models environment effects nearly as well as with PCs (*Figure 8*), consistent with previous findings (*Vilhjálmsson and Nordborg, 2013*; *Wang et al., 2022*) and with environment inflating heritability estimates using LMM (*Heckerman et al., 2016*). Moreover, modeling the true environment groups as fixed categorical effects always substantially improved $AUC_{PR}$ compared to modeling them with PCs (*Figure 8*, *Table 6*). Modeling numerous environment groups as fixed effects does result in deflated p-values (*Figure 8*, *Table 6*), which we expect would be avoided by modeling them as random effects, a strategy we chose not to

**Table 6.** Overview of PCA and LMM evaluations for environment simulations.

| Dataset | Metric | Trait* | Cal.† | $r$‡ | p-value § | $r$‡ | Cal.† | p-value § | Best ¶ | Cal.† | p-value § | Best ¶ |
|---|---|---|---|---|---|---|---|---|---|---|---|---|
| | | | LMM $r = 0$ vs best $r$ | | | PCA vs LMM $r = 0$ | | | | LMM lab. $r = 0$ vs PCA/LMM | | |
| Admix. Large sim. | $|\mathrm{SRMSD}_p|$ | FES | True | 0 | 1 | 83 | True | 0.38 | Tie | True | 1.8e-14* | PCA/LMM |
| Admix. Small sim. | $|\mathrm{SRMSD}_p|$ | FES | True | 0 | 1 | 90 | True | 0.001 | Tie | False | 1.4e-14* | PCA/LMM |
| Admix. Family sim. | $|\mathrm{SRMSD}_p|$ | FES | True | 4 | 0.18 | 90 | False | 3.9e-10* | LMM | True | 0.066 | LMM/LMM lab. |
| Human Origins | $|\mathrm{SRMSD}_p|$ | FES | True | 9 | 3.9e-05* | 90 | False | 1.4e-08* | LMM | False | 3.9e-10* | LMM |
| HGDP | $|\mathrm{SRMSD}_p|$ | FES | True | 0 | 1 | 90 | True | 0.0037 | Tie | False | 2.1e-09* | PCA/LMM |
| 1000 Genomes | $|\mathrm{SRMSD}_p|$ | FES | False | 8 | 8.8e-08* | 85 | True | 0.053 | Tie | True | 3.9e-10* | LMM lab. |
| Admix. Large sim. | $|\mathrm{SRMSD}_p|$ | RC | True | 0 | 1 | 60 | True | 0.033 | Tie | True | 6.3e-10* | PCA/LMM |
| Admix. Small sim. | $|\mathrm{SRMSD}_p|$ | RC | True | 0 | 1 | 9 | True | 0.85 | Tie | False | 1.4e-14* | PCA/LMM |
| Admix. Family sim. | $|\mathrm{SRMSD}_p|$ | RC | True | 5 | 0.14 | 90 | False | 3.9e-10* | LMM | True | 0.011 | LMM/LMM lab. |
| Human Origins | $|\mathrm{SRMSD}_p|$ | RC | False | 9 | 1.1e-08* | 90 | True | 2.3e-07* | PCA | False | 3.9e-10* | PCA |
| HGDP | $|\mathrm{SRMSD}_p|$ | RC | True | 0 | 1 | 89 | True | 6.5e-09* | PCA | False | 3.9e-10* | PCA |
| 1000 Genomes | $|\mathrm{SRMSD}_p|$ | RC | False | 8 | 1.6e-08* | 88 | True | 4.9e-09* | PCA | True | 0.09 | PCA/LMM lab. |
| Admix. Large sim. | $\mathrm{AUC_{PR}}$ | FES | | 4 | 2.4e-06* | 6 | | 0.0021 | Tie | | 1.8e-15* | LMM lab. |
| Admix. Small sim. | $\mathrm{AUC_{PR}}$ | FES | | 3 | 0.055 | 4 | | 0.033 | Tie | | 0.28 | Tie |
| Admix. Family sim. | $\mathrm{AUC_{PR}}$ | FES | | 12 | 7e-04 | 63 | | 3.9e-10* | LMM | | 3.9e-10* | LMM lab. |
| Human Origins | $\mathrm{AUC_{PR}}$ | FES | | 20 | 3.7e-06* | 90 | | 1.4e-05* | LMM | | 3.9e-10* | LMM lab. |
| HGDP | $\mathrm{AUC_{PR}}$ | FES | | 12 | 4.3e-06* | 45 | | 0.0044 | Tie | | 3.9e-10* | LMM lab. |
| 1000 Genomes | $\mathrm{AUC_{PR}}$ | FES | | 9 | 1.9e-08* | 55 | | 0.028 | Tie | | 3.9e-10* | LMM lab. |
| Admix. Large sim. | $\mathrm{AUC_{PR}}$ | RC | | 4 | 0.00085 | 5 | | 0.0018 | Tie | | 5e-10* | LMM lab. |
| Admix. Small sim. | $\mathrm{AUC_{PR}}$ | RC | | 2 | 0.13 | 5 | | 0.093 | Tie | | 0.0028 | Tie |
| Admix. Family sim. | $\mathrm{AUC_{PR}}$ | RC | | 9 | 0.01 | 86 | | 1.7e-09* | LMM | | 3.9e-10* | LMM lab. |
| Human Origins | $\mathrm{AUC_{PR}}$ | RC | | 22 | 0.0039 | 90 | | 1e-06* | PCA | | 3.9e-10* | LMM lab. |
| HGDP | $\mathrm{AUC_{PR}}$ | RC | | 19 | 0.0057 | 64 | | 2.8e-05* | PCA | | 3e-07* | LMM lab. |
| 1000 Genomes | $\mathrm{AUC_{PR}}$ | RC | | 9 | 8.7e-05* | 87 | | 1.2e-09* | PCA | | 4.4e-10* | LMM lab. |

*FES: Fixed Effect Sizes, RC: Random Coefficients.

†Calibrated: whether mean $|\mathrm{SRMSD}_p| < 0.01$ over 50 replicates.

‡Value of $r$ (number of PCs) with minimum mean $|\mathrm{SRMSD}_p|$ or maximum mean $\mathrm{AUC_{PR}}$.

§Wilcoxon paired 1-tailed test of distributions ($|\mathrm{SRMSD}_p|$ or $\mathrm{AUC_{PR}}$) between models in header. Asterisk marks significant value using Bonferroni threshold ($p < \alpha/n_{\text{tests}}$ with $\alpha = 0.01$ and $n_{\text{tests}} = 72$ is the number of tests in this table).

¶Tie if no significant difference using Bonferroni threshold; in last column, pairwise ties are specified and "Tie" is three-way tie.

pursue here as it is both a circular evaluation (the true effects were drawn from that model) and out of scope. Overall, including PCs to model environment effects yields limited power gains if at all, even in an LMM, and is no replacement for more adequate modeling of environment whenever possible.

Previous studies found that PCA was better calibrated than LMM for unusually differentiated markers (*Price et al., 2010*; *Wu et al., 2011*; *Yang et al., 2014*), which as simulated were an artificial scenario not based on a population genetics model, and are otherwise believed to be unusual (*Sul and Eskin, 2013*; *Price et al., 2013*). Our evaluations on real human data, which contain such loci in

relevant proportions if they exist, do not replicate that result. Family relatedness strongly favors LMM, an advantage that probably outweighs this potential PCA benefit in real data.

Relative to LMM, the behavior of PCA fell between two extremes. When PCA performed well, there was a small number of PCs with both calibrated p-values and $AUC_{PR}$ near that of LMM without PCs. Conversely, PCA performed poorly when no number of PCs had either calibrated p-values or acceptably large $AUC_{PR}$. There were no cases where high numbers of PCs optimized an acceptable $AUC_{PR}$, or cases with miscalibrated p-values but high $AUC_{PR}$. PCA performed well in the admixture simulations (without families, both trait models), real human genotypes with RC traits, and the subpopulation tree simulations (both trait models). Conversely, PCA performed poorly in the admixed family simulation (both trait models) and the real human genotypes with FES traits.

PCA assumes that genetic relatedness is restricted to a low-dimensional subspace, whereas LMM can handle high-dimensional relatedness. Thus, PCA performs well in the admixture simulation, which is explicitly low-dimensional (see Genotype simulation from the admixture model), and our subpopulation tree simulations, which are likely well approximated by a few dimensions despite the large number of subpopulations because there are few long branches. Conversely, PCA performs poorly under family structure because its kinship matrix is high-dimensional (*Figure 6—figure supplement 1*). However, estimating the latent space dimensions of real datasets is challenging because estimated eigenvalues have biased distributions (*Hayashi et al., 2018*). Kinship matrix rank estimated using the Tracy-Widom test (*Patterson et al., 2006*) did not fully predict the datasets that PCA performs well on. In contrast, estimated local kinship finds considerable cryptic family relatedness in all real human datasets and better explains why PCA performs poorly there. The trait model also influences the relative performance of PCA, so genotype-only parameters (eigenvalues or local kinship) alone cannot tell the full story. There are related tests for numbers of dimensions that consider the trait which we did not consider, including the Bayesian information criterion for the regression with PCs against the trait (*Zhu and Yu, 2009*). Additionally, PCA and LMM goodness of fit could be compared using the coefficient of determination generalized for LMMs (*Sun et al., 2010*).

PCA is at best underpowered relative to LMMs, and at worst miscalibrated regardless of the numbers of PCs included, in real human genotype tests. Among our simulations, such poor performance occurred only in the admixed family. Local kinship estimates reveal considerable family relatedness in the real datasets absent in the corresponding subpopulation tree simulations. Admixture is also absent in our tree simulations, but our simulations and theory show that admixture is well handled by PCA. Hundreds of close relative pairs have been identified in 1000 Genomes (*Gazal et al., 2015*; *Al Khudhair et al., 2015*; *Fedorova et al., 2016*; *Schlauch et al., 2017*), but their removal does not improve PCA performance sufficiently in our tests, so the larger number of more distantly related pairs are PCA's most serious obstacle in practice. Distant relatives are expected to be numerous in any large human dataset (*Henn et al., 2012*; *Shchur and Nielsen, 2018*; *Loh et al., 2018*). Our FES trait tests show that family relatedness is more challenging when rarer variants have larger coefficients. Overall, the high relatedness dimensions induced by family relatedness is the key challenge for PCA association in modern datasets that is readily overcome by LMM.

Our tests also found PCA robust to large numbers of PCs, far beyond the optimal choice, agreeing with previous anecdotal observations (*Price et al., 2006*; *Kang et al., 2010*), in contrast to using too few PCs for which there is a large performance penalty. The exception was the small sample size simulation, where only small numbers of PCs performed well. In contrast, LMM is simpler since there is no need to choose the number of PCs. However, an LMM with a large number of covariates may have conservative p-values, as observed for LMM with large numbers of PCs, which is a weakness of the score test used by the LMM we evaluated that may be overcome with other statistical tests. Simulations or post hoc evaluations remain crucial for ensuring that statistics are calibrated.

There are several variants of the PCA and LMM analyses, most designed for better modeling linkage disequilibrium (LD), that we did not evaluate directly, in which PCs are no longer exactly the top eigenvectors of the kinship matrix (if estimated with different approaches), although this is not a crucial aspect of our arguments. We do not consider the case where samples are projected onto PCs estimated from an external sample (*Privé et al., 2020*), which is uncommon in association studies, and whose primary effect is shrinkage, so if all samples are projected then they are all equally affected and larger regression coefficients compensate for the shrinkage, although this will no longer be the case if only a portion of the sample is projected onto the PCs of the rest of the sample. Another

approach tests PCs for association against every locus in the genome in order to identify and exclude PCs that capture LD structure (which is localized) instead of ancestry (which should be present across the genome; *Privé et al., 2020*); a previous proposal removes LD using an autocorrelation model prior to estimating PCs (*Patterson et al., 2006*). These improved PCs remain inadequate models of family relatedness, so an LMM will continue to outperform them in that setting. Similarly, the leave-one-chromosome-out (LOCO) approach for estimating kinship matrices for LMMs prevents the test locus and loci in LD with it from being modeled by the random effect as well, which is called 'proximal contamination' (*Lippert et al., 2011*; *Yang et al., 2014*). While LOCO kinship estimates vary for each chromosome, they continue to model family relatedness, thus maintaining their key advantage over PCA. The LDAK model estimates kinship instead by weighing loci taking LD into account (*Speed et al., 2012*). LD effects must be adjusted for, if present, so in unfiltered data we advise the previous methods be applied. However, in this work, simulated genotypes do not have LD, and the real data-sets were filtered to remove LD, so here there is no proximal contamination and LD confounding is minimized if present at all, so these evaluations may be considered the ideal situation where LD effects have been adjusted successfully, and in this setting LMM outperforms PCA. Overall, these alternative PCs or kinship matrices differ from their basic counterparts by either the extent to which LD influences the estimates (which may be a confounder in a small portion of the genome, by defini-tion) or by sampling noise, neither of which are expected to change our key conclusion.

One of the limitations of this work include relatively small sample sizes compared to modern asso-ciation studies. However, our conclusions are not expected to change with larger sample sizes, as cryptic family relatedness will continue to be abundant in such data, if not increase in abundance, and thus give LMMs an advantage over PCA (*Henn et al., 2012*; *Shchur and Nielsen, 2018*; *Loh et al., 2018*). One reason PCA has been favored over classic LMMs is because PCA's runtime scales much better with increasing sample size. However, recent approaches not tested in this work have made LMMs more scalable and applicable to biobank-scale data (*Loh et al., 2015*; *Zhou et al., 2018*; *Mbatchou et al., 2021*), so one clear next step is carefully evaluating these approaches in simulations with larger sample sizes. A different benefit for including PCs were recently reported for BOLT-LMM, which does not result in greater power but rather in reduced runtime, a property that may be specific to its use of scalable algorithms such as conjugate gradient and variational Bayes (*Loh et al., 2018*). Many of these newer LMMs also no longer follow the infinitesimal model of the basic LMM (*Loh et al., 2015*; *Mbatchou et al., 2021*), and employ novel approximations, which are features not evaluated in this work and worthy of future study.

Another limitation of this work is ignoring rare variants, a necessity given our smaller sample sizes, where rare variant association is miscalibrated and underpowered. Using simulations mimicking the UK Biobank, recent work has found that rare variants can have a more pronounced structure than common variants, and that modeling this rare variant structure (with either PCA and LMM) may better model environment confounding, reduce inflation in association studies, and ameliorate stratification in polygenic risk scores (*Zaidi and Mathieson, 2020*). Better modeling rare variants and their structure is a key next step in association studies.

The largest limitation of our work is that we only considered quantitative traits. Previous evaluations involving case-control traits tended to report PCA-LMM ties or mixed results, an observation poten-tially confounded by the use of low-dimensional simulations without family relatedness (*Table 1*). An additional concern is case-control ascertainment bias and imbalance, which appears to affect LMMs more severely, although recent work appears to solve this problem (*Yang et al., 2014*; *Zhou et al., 2018*). Future evaluations should aim to include our simulations and real datasets, to ensure that previous results were not biased in favor of PCA by not simulating family structure or larger coeffi-cients for rare variants that are expected for diseases by various selection models.

Overall, our results lead us to recommend LMM over PCA for association studies in general. Although PCA offer flexibility and speed compared to LMM, additional work is required to ensure that PCA is adequate, including removal of close relatives (lowering sample size and wasting resources) followed by simulations or other evaluations of statistics, and even then PCA may perform poorly in terms of both type I error control and power. The large numbers of distant relatives expected of any real dataset all but ensures that PCA will perform poorly compared to LMM (*Henn et al., 2012*; *Shchur and Nielsen, 2018*; *Loh et al., 2018*). Our findings also suggest that related applications such as polygenic models may enjoy gains in power and accuracy by employing an LMM instead of PCA

to model relatedness (*Rakitsch et al., 2013*; *Qian et al., 2020*). PCA remains indispensable across population genetics, from visualizing population structure and performing quality control to its deep connection to admixture models, but the time has come to limit its use in association testing in favor of LMM or other, richer models capable of modeling all forms of relatedness.

## Materials and methods

### The complex trait model and PCA and LMM approximations

Let $x_{ij} \in \{0, 1, 2\}$ be the genotype at the biallelic locus $i$ for individual $j$, which counts the number of reference alleles. Suppose there are $n$ individuals and $m$ loci, $\mathbf{X} = (x_{ij})$ is their $m \times n$ genotype matrix, and $\mathbf{y}$ is the length-$n$ column vector of individual trait values. The additive linear model for a quantitative (continuous) trait is:

$$\mathbf{y} = \mathbf{1}\alpha + \mathbf{X}'\boldsymbol{\beta} + \mathbf{Z}'\boldsymbol{\eta} + \boldsymbol{\epsilon}, \qquad (1)$$

where 1 is a length-$n$ vector of ones, $\alpha$ is the scalar intercept coefficient, $\boldsymbol{\beta}$ is the length-$m$ vector of locus coefficients, $\mathbf{Z}$ is a design matrix of environment effects and other covariates, $\boldsymbol{\eta}$ is the vector of environment coefficients, $\boldsymbol{\epsilon}$ is a length-$n$ vector of residuals, and the superscript prime symbol ($'$) denotes matrix transposition. The residuals follow $\epsilon_j \sim \text{Normal}(0, \sigma_\epsilon^2)$ independently per individual $j$, for some $\sigma_\epsilon^2$.

The full model of *Equation 1*, which has a coefficient for each of the $m$ loci, is underdetermined in current datasets where $m \gg n$. The PCA and LMM models, respectively, approximate the full model fit at a single locus $i$:

$$\text{PCA:} \quad \mathbf{y} = \mathbf{1}\alpha + \mathbf{x}_i\beta_i + \mathbf{U}_r\boldsymbol{\gamma}_r + \mathbf{Z}'\boldsymbol{\eta} + \boldsymbol{\epsilon}, \qquad (2)$$

$$\text{LMM:} \quad \mathbf{y} = \mathbf{1}\alpha + \mathbf{x}_i\beta_i + \mathbf{s} + \mathbf{Z}'\boldsymbol{\eta} + \boldsymbol{\epsilon}, \qquad \mathbf{s} \sim \text{Normal}\left(\mathbf{0}, 2\sigma_s^2\boldsymbol{\Phi}^T\right), \qquad (3)$$

where $\mathbf{x}_i$ is the length-$n$ vector of genotypes at locus $i$ only, $\beta_i$ is the locus coefficient, $\mathbf{U}_r$ is an $n \times r$ matrix of PCs, $\boldsymbol{\gamma}_r$ is the length-$r$ vector of PC coefficients, $\mathbf{s}$ is a length-$n$ vector of random effects, $\boldsymbol{\Phi}^T = (\varphi_{jk}^T)$ is the $n \times n$ kinship matrix conditioned on the ancestral population $T$, and $\sigma_s^2$ is a variance factor. Both models condition the regression of the focal locus $i$ on an approximation of the total polygenic effect $\mathbf{X}'\boldsymbol{\beta}$ with the same covariance structure, which is parameterized by the kinship matrix. Under the kinship model, genotypes are random variables obeying

$$\text{E}[\mathbf{x}_i|T] = 2p_i^T\mathbf{1}, \qquad \text{Cov}(\mathbf{x}_i|T) = 4p_i^T(1 - p_i^T)\boldsymbol{\Phi}^T, \qquad (4)$$

where $p_i^T$ is the ancestral allele frequency of locus $i$ (*Malécot, 1948*; *Wright, 1949*; *Jacquard, 1970*; *Astle and Balding, 2009*). Assuming independent loci, the covariance of the polygenic effect is

$$\text{Cov}(\mathbf{X}'\boldsymbol{\beta}) = 2\sigma_s^2\boldsymbol{\Phi}^T, \qquad \sigma_s^2 = \sum_{i=1}^{m} 2p_i^T(1 - p_i^T)\beta_i^2,$$

which is readily modeled by the LMM random effect $\mathbf{s}$, where the difference in mean is absorbed by the intercept. Alternatively, consider the eigendecomposition of the kinship matrix $\boldsymbol{\Phi}^T = \mathbf{U}\boldsymbol{\Lambda}\mathbf{U}'$ where $\mathbf{U}$ is the $n \times n$ eigenvector matrix and $\boldsymbol{\Lambda}$ is the $n \times n$ diagonal matrix of eigenvalues. The random effect can be written as

$$\mathbf{s} = \mathbf{U}\boldsymbol{\gamma}_{\text{LMM}}, \qquad \boldsymbol{\gamma}_{\text{LMM}} \sim \text{Normal}(\mathbf{0}, 2\sigma_s^2\boldsymbol{\Lambda}),$$

which follows from the affine transformation property of multivariate normal distributions. Therefore, the PCA term $\mathbf{U}_r\boldsymbol{\gamma}_r$ can be derived from the above equation under the additional assumption that the kinship matrix has approximate rank $r$ and the coefficients $\boldsymbol{\gamma}_r$ are fit without constraints. In contrast, the LMM uses all eigenvectors, while effectively shrinking their coefficients $\boldsymbol{\gamma}_{\text{LMM}}$ as all random effects models do, although these parameters are marginalized (*Astle and Balding, 2009*; *Janss et al., 2012*; *Hoffman and Dubé, 2013*; *Zhang and Pan, 2015*). PCA has more parameters than LMM, so it may overfit more: ignoring the shared terms in *Equation 2* and *Equation 3*, PCA fits $r$ parameters (length of $\boldsymbol{\gamma}$), whereas LMMs fit only one ($\sigma_s^2$).

In practice, the kinship matrix used for PCA and LMM is estimated with variations of a method-of-moments formula applied to standardized genotypes $\mathbf{X}_S$, which is derived from *Equation 4*:

$$\mathbf{X}_S = \left( \frac{x_{ij} - 2\hat{p}_i^T}{\sqrt{4\hat{p}_i^T \left(1 - \hat{p}_i^T\right)}} \right), \qquad \hat{\Phi}^T = \frac{1}{m}\mathbf{X}_S'\mathbf{X}_S, \tag{5}$$

where the unknown $p_i^T$ is estimated by $\hat{p}_i^T = \frac{1}{2n}\sum_{j=1}^n x_{ij}$ (*Price et al., 2006*; *Kang et al., 2008*; *Kang et al., 2010*; *Yang et al., 2011*; *Zhou and Stephens, 2012*; *Yang et al., 2014*; *Loh et al., 2015*; *Sul et al., 2018*; *Zhou et al., 2018*). However, this kinship estimator has a complex bias that differs for every individual pair, which arises due to the use of this estimated $\hat{p}_i^T$ (*Ochoa and Storey, 2021*; *Ochoa and Storey, 2019*). Nevertheless, in PCA and LMM these biased estimates perform as well as unbiased ones (*Hou et al., 2023b*).

We selected fast and robust software implementing the basic PCA and LMM models. PCA association was performed with plink2 (*Chang et al., 2015*). The quantitative trait association model is a linear regression with covariates, evaluated using the t-test. PCs were calculated with plink2, which equal the top eigenvectors of *Equation 5* after removing loci with minor allele frequency MAF < 0.1.

LMM association was performed using GCTA (*Yang et al., 2011*; *Yang et al., 2014*). Its kinship estimator equals *Equation 5*. PCs were calculated using GCTA from its kinship estimate. Association significance is evaluated with a score test. In the small simulation only, GCTA with large numbers of PCs had convergence and singularity errors in some replicates, which were treated as missing data.

## Simulations

Every simulation was replicated 50 times, drawing anew all genotypes (except for real datasets) and traits. Below we use the notation $f_A^B$ for the inbreeding coefficient of a subpopulation $A$ from another subpopulation $B$ ancestral to $A$. In the special case of the *total* inbreeding of $A$, $f_A^T$, $T$ is an overall ancestral population, which is ancestral to every individual under consideration, such as the most recent common ancestor (MRCA) population.

### Genotype simulation from the admixture model

The basic admixture model is as described previously (*Ochoa and Storey, 2021*) and is implemented in the R package bnpsd. Both Large and Family simulations have $n = 1,000$ individuals, while Small has $n = 100$. The number of loci is $m = 100,000$. Individuals are admixed from $K = 10$ intermediate subpopulations, or ancestries. Each subpopulation $S_u$ ($u \in \{1, ..., K\}$) is at coordinate $u$ and has an inbreeding coefficient $f_{S_u}^T = u\tau$ for some $\tau$. Ancestry proportions $q_{ju}$ for individual $j$ and $S_u$ arise from a random walk with spread $\sigma$ on the 1D geography, and $\tau$ and $\sigma$ are fit to give $F_{\mathrm{ST}} = 0.1$ and mean kinship $\bar{\theta}^T = 0.5F_{\mathrm{ST}}$ for the admixed individuals (*Ochoa and Storey, 2021*). Random ancestral allele frequencies $p_i^T$, subpopulation allele frequencies $p_i^{S_u}$, individual-specific allele frequencies $\pi_{ij}$, and genotypes $x_{ij}$ are drawn from this hierarchical model:

$$p_i^T \sim \mathrm{Uniform}(0.01, 0.5),$$

$$p_i^{S_u}|p_i^T \sim \mathrm{Beta}\left(p_i^T\left(\frac{1}{f_{S_u}^T} - 1\right), \left(1 - p_i^T\right)\left(\frac{1}{f_{S_u}^T} - 1\right)\right),$$

$$\pi_{ij} = \sum_{u=1}^K q_{ju}p_i^{S_u},$$

$$x_{ij}|\pi_{ij} \sim \mathrm{Binomial}(2, \pi_{ij}),$$

where this Beta is the Balding-Nichols distribution (*Balding and Nichols, 1995*) with mean $p_i^T$ and variance $p_i^T\left(1 - p_i^T\right)f_{S_u}^T$. Fixed loci ($i$ where $x_{ij} = 0$ for all $j$, or $x_{ij} = 2$ for all $j$) are drawn again from the model, starting from $p_i^T$, iterating until no loci are fixed. Each replicate draws a genotypes starting from $p_i^T$.

As a brief aside, we prove that global ancestry proportions as covariates is equivalent in expectation to using PCs under the admixture model. Note that the latent space of $\mathbf{X}$, which is the subspace to which the data is constrained by the admixture model, is given by $(\pi_{ij})$, which has $K$ dimensions

(number of columns of $\mathbf{Q} = (q_{ju})$), so the top $K$ PCs span this space. Since associations include an intercept term ($\mathbf{1}\alpha$ in *Equation 2*), estimated PCs are orthogonal to 1 (note $\hat{\Phi}^T\mathbf{1} = \mathbf{0}$ because $\mathbf{X}_S\mathbf{1} = \mathbf{0}$), and the sum of rows of $\mathbf{Q}$ sums to one, then only $K - 1$ PCs plus the intercept are needed to span the latent space of this admixture model.

## Genotype simulation from random admixed families

We simulated a pedigree with admixed founders, no close relative pairings, assortative mating based on a 1D geography (to preserve admixture structure), random family sizes, and arbitrary numbers of generations (20 here). This simulation is implemented in the R package simfam. Generations are drawn iteratively. Generation 1 has $n = 1000$ individuals from the above admixture simulation ordered by their 1D geography. Local kinship measures pedigree relatedness; in the first generation, everybody is locally unrelated and outbred. Individuals are randomly assigned sex. In the next generation, individuals are paired iteratively, removing random males from the pool of available males and pairing them with the nearest available female with local kinship $< 1/4^3$ (stay unpaired if there are no matches), until there are no more available males or females. Let $n = 1000$ be the desired population size, $n_m = 1$ the minimum number of children per family and $n_f$ the number of families (paired parents) in the current generation, then the number of additional children (beyond the minimum) is drawn from Poisson($n/n_f - n_m$). Let $\delta$ be the difference between desired and current population sizes. If $\delta > 0$, then $\delta$ random families are incremented by 1. If $\delta < 0$, then $|\delta|$ random families with at least $n_m + 1$ children are decremented by 1. If $|\delta|$ exceeds the number of families, all families are incremented or decremented as needed and the process is iterated. Children are assigned sex randomly, and are reordered by the average coordinate of their parents. Children draw alleles from their parents independently per locus. A new random pedigree is drawn for each replicate, as well as new founder genotypes from the admixture model.

## Genotype simulation from a subpopulation tree model

This model draws subpopulations allele frequencies from a hierarchical model parameterized by a tree, which is also implemented in bnpsd and relies on the R package ape for general tree data structures and methods (*Paradis and Schliep, 2019*). The ancestral population $T$ is the root, and each node is a subpopulation $S_w$ indexed arbitrarily. Each edge between $S_w$ and its parent population $P_w$ has an inbreeding coefficient $f^{P_w}_{S_w}$. $P^T_i$ are drawn from a given distribution, which is constructed to mimic each real dataset in Appendix 1. Given the allele frequencies $p^{P_w}_i$ of the parent population, $S_w$'s allele frequencies are drawn from:

$$p^{S_w}_i | p^{P_w}_i \sim \text{Beta}\left(p^{P_w}_i\left(\frac{1}{f^{P_w}_{S_w}} - 1\right), \left(1 - p^{P_w}_i\right)\left(\frac{1}{f^{P_w}_{S_w}} - 1\right)\right).$$

Individuals $j$ in $S_w$ draw genotypes from its allele frequency: $x_{ij}|p^{S_w}_i \sim \text{Binomial}\left(2, p^{S_w}_i\right)$. Loci with MAF $< 0.01$ are drawn again starting from the $p^T_i$ distribution, iterating until no such loci remain.

## Fitting subpopulation tree to real data

We developed new methods to fit trees to real data based on unbiased kinship estimates from popkin, implemented in bnpsd. A tree with given inbreeding coefficients $f^{P_w}_{S_w}$ for its edges (between subpopulation $S_w$ and its parent $P_w$) gives rise to a coancestry matrix $\vartheta^T_{uv}$ for a subpopulation pair $(S_u, S_v)$, and the goal is to recover these edge inbreeding coefficients from coancestry estimates. Coancestry values are total inbreeding coefficients of the MRCA population of each subpopulation pair. Therefore, we calculate $\hat{f}^T_{S_w}$ for every $S_w$ recursively from the root as follows. Nodes with parent $P_w = T$ are already as desired. Given $\hat{f}^T_{P_w}$, the desired $\hat{f}^T_{S_w}$ is calculated via the 'additive edge' $\delta_w$ (*Ochoa and Storey, 2021*):

$$f^T_{S_w} = f^T_{P_w} + \delta_w, \qquad \delta_w = f^{P_w}_{S_w}\left(1 - f^T_{P_w}\right). \tag{6}$$

These $\delta_w \geq 0$ because $0 \leq f_{S_w}^{P_w}, f_{P_w}^{T} \leq 1$ for every $w$. Edge inbreeding coefficients can be recovered from additive edges: $f_{S_w}^{P_w} = \delta_w/(1 - f_{P_w}^{T})$. Overall, coancestry values are sums of $\delta_w$ over common ancestor nodes,

$$\vartheta_{uv}^{T} = \sum_{w} \delta_w I_w(u, v),$$

(7)

where the sum includes all $w$, and $I_w(u, v)$ equals 1 if $S_w$ is a common ancestor of $S_u, S_v$, 0 otherwise. Note that $I_w(u, v)$ reflects tree topology and $\delta_w$ edge values.

To estimate population-level coancestry, first kinship ($\hat{\varphi}_{jk}^{T}$) is estimated using popkin (**Ochoa and Storey, 2021**). Individual coancestry ($\hat{\theta}_{jk}^{T}$) is estimated from kinship using

$$\hat{\theta}_{jk}^{T} = \begin{cases} \hat{\varphi}_{jk}^{T} & \text{if} \quad k \neq j, \\ \hat{f}_{j}^{T} = 2\hat{\varphi}_{jj}^{T} - 1 & \text{if} \quad k = j. \end{cases}$$

(8)

Lastly, coancestry $\hat{\vartheta}_{uv}^{T}$ between subpopulations are averages of individual coancestry values:

$$\hat{\vartheta}_{uv}^{T} = \frac{1}{|S_u||S_v|} \sum_{j \in S_u} \sum_{k \in S_v} \hat{\theta}_{jk}^{T}.$$

Topology is estimated with hierarchical clustering using the weighted pair group method with arithmetic mean (**Sokal and Michener, 1958**), with distance function $d(S_u, S_v) = \max\left\{\hat{\vartheta}_{uv}^{T}\right\} - \hat{\vartheta}_{uv}^{T}$, which succeeds due to the monotonic relationship between node depth and coancestry (**Equation 7**). This algorithm recovers the true topology from the true coancestry values, and performs well for estimates from genotypes.

To estimate tree edge lengths, first $\delta_w$ are estimated from $\hat{\vartheta}_{uv}^{T}$ and the topology using **Equation 7** and non-negative least squares linear regression (**Lawson and Hanson, 1974**) (implemented in nnls; **Mullen, 2012**) to yield non-negative $\delta_w$, and $f_{S_w}^{P_w}$ are calculated from $\delta_w$ by reversing **Equation 5**. To account for small biases in coancestry estimation, an intercept term $\delta_0$ is included ($I_0(u, v) = 1$ for all $u, v$), and when converting $\delta_w$ to $f_{S_w}^{P_w}$, $\delta_0$ is treated as an additional edge to the root, but is ignored when drawing allele frequencies from the tree.

### Trait simulation

Traits are simulated from the quantitative trait model of **Equation 1**, with novel bias corrections for simulating the desired heritability from real data relying on the unbiased kinship estimator popkin (**Ochoa and Storey, 2021**). This simulation is implemented in the R package simtrait. All simulations have a fixed narrow-sense heritability of $h^2$, a variance proportion due to environment effects $\sigma_\eta^2$, and residuals are drawn from $\epsilon_j \sim \text{Normal}(0, \sigma_\epsilon^2)$ with $\sigma_\epsilon^2 = 1 - h^2 - \sigma_\eta^2$. The number of causal loci $m_1$, which determines the average coefficient size, is chosen with the heuristic formula $m_1 = \text{round}(nh^2/8)$, which empirically balances power well with varying $n$ and $h^2$. The set of causal loci $C$ is drawn anew for each replicate, from loci with MAF $\geq 0.01$ to avoid rare causal variants, which are not discoverable by PCA or LMM at the sample sizes we considered. Letting $v_i^{T} = p_i^{T}\left(1 - p_i^{T}\right)$, the effect size of locus $i$ equals $2v_i^{T}\beta_i^2$, its contribution of the trait variance (**Park et al., 2010**). Under the *fixed effect sizes* (FES) model, initial causal coefficients are

$$\beta_i = \frac{1}{\sqrt{2v_i^{T}}}$$

for known $p_i^{T}$; otherwise $v_i^{T}$ is replaced by the unbiased estimator (**Ochoa and Storey, 2021**) $\hat{v}_i^{T} = \hat{p}_i^{T}\left(1 - \hat{p}_i^{T}\right)/(1 - \bar{\varphi}^{T})$, where $\bar{\varphi}^{T}$ is the mean kinship estimated with popkin. Each causal locus is multiplied by –1 with probability 0.5. Alternatively, under the *random coefficients* (RC) model, initial causal coefficients are drawn independently from $\beta_i \sim \text{Normal}(0, 1)$. For both models, the initial genetic variance is $\sigma_0^2 = \sum_{i \in C} 2v_i^{T}\beta_i^2$, replacing $v_i^{T}$ with $\hat{v}_i^{T}$ for unknown $p_i^{T}$ (so $\sigma_0^2$ is an unbiased estimate), so

**Table 7.** Variance parameters of trait simulations.

| Trait variance type | $h^2$ | $\sigma_\eta^2$ | $\sigma_\epsilon^2$ |
| --- | --- | --- | --- |
| High heritability | 0.8 | 0.0 | 0.2 |
| Low heritability | 0.3 | 0.0 | 0.7 |
| Environment | 0.3 | 0.5 | 0.2 |

we multiply every initial $\beta_i$ by $\frac{h}{\sigma_0}$ to have the desired heritability. Lastly, for known $p_i^T$, the intercept coefficient is $\alpha = -\sum_{i \in C} 2p_i^T \beta_i$. When $p_i^T$ are unknown, $\hat{p}_i^T$ should not replace $p_i^T$ since that distorts the trait covariance (for the same reason the standard kinship estimator in **Equation 5** is biased), which is avoided with

$$\alpha = -\frac{2}{m_1} \left( \sum_{i \in C} \hat{p}_i^T \right) \left( \sum_{i \in C} \beta_i \right).$$

Simulations optionally included multiple environment group effects, similarly to previous models (**Zhang and Pan, 2015**; **Wang et al., 2022**), as follows. Each independent environment $i$ has predefined groups, and each group $g$ has random coefficients drawn independent from $\eta_{gi} \sim \text{Normal}(0, \sigma_{\eta i}^2)$ where $\sigma_{\eta i}^2$ is a specified variance proportion for environment $i$. **Z** has individuals along columns and environment-groups along rows, and it contains indicator variables: 1 if the individual belongs to the environment-group, 0 otherwise.

We performed trait simulations with the following variance parameters (**Table 7**): *high heritability* used $h^2 = 0.8$ and no environment effects; *low heritability* used $h^2 = 0.3$ and no environment effects; lastly, *environment* used $h^2 = 0.3, \sigma_{\eta 1}^2 = 0.3, \sigma_{\eta 2}^2 = 0.2$ (total $\sigma_\eta^2 = \sigma_{\eta 1}^2 + \sigma_{\eta 2}^2 = 0.5$). For real genotype datasets, the groups are the continental (environment 1) and fine-grained (environment 2) subpopulation labels given (see next subsection). For simulated genotypes, we created these labels by grouping by the index $j$ (geographical coordinate) of each simulated individual, assigning group $g = \text{ceiling}(jk_i/n)$ where $k_i$ is the number of groups in environment $i$, and we selected $k_1 = 5$ and $k_2 = 25$ to mimic the number of groups in each level of 1000 Genomes (**Table 2**).

## Real human genotype datasets

The three datasets were processed as before (**Ochoa and Storey, 2019**; summarized below), except with an additional filter so loci are in approximate linkage equilibrium and rare variants are removed. All processing was performed with plink2 (**Chang et al., 2015**), and analysis was uniquely enabled by the R packages BEDMatrix (**Grueneberg and de Los Campos, 2019**) and genio. Each dataset groups individuals in a two-level hierarchy: continental and fine-grained subpopulations. Final dataset sizes are in **Table 2**.

We obtained the full (including non-public) Human Origins by contacting the authors and agreeing to their usage restrictions. The Pacific data (**Skoglund et al., 2016**) was obtained separately from the rest (**Lazaridis et al., 2014**; **Lazaridis et al., 2016**), and datasets were merged using the intersection of loci. We removed ancient individuals, and individuals from singleton and non-native subpopulations. Non-autosomal loci were removed. Our analysis of both the whole-genome sequencing (WGS) version of HGDP (**Bergström et al., 2020**) and the high-coverage NYGC version of 1000 Genomes (**Fairley et al., 2020**) was restricted to autosomal biallelic SNP loci with filter "PASS".

Since our evaluations assume uncorrelated loci, we filtered each real dataset with plink2 using parameters "--indep-pairwise 1000kb 0.3", which iteratively removes loci that have a greater than 0.3 squared correlation coefficient with another locus that is within 1000 kb, stopping until no such loci remain. Since all real datasets have numerous rare variants, while PCA and LMM are not able to detect associations involving rare variants, we removed all loci with $\text{MAF} < 0.01$. Lastly, only HGDP had loci with over 10% missingness removed, as they were otherwise 17% of remaining loci (for Human Origins and 1000 Genomes they were under 1% of loci so they were not removed). Kinship matrix rank and eigenvalues were calculated from popkin kinship estimates. Eigenvalues were assigned p-values with twstats of the Eigensoft package (**Patterson et al., 2006**), and kinship matrix rank was estimated as

the largest number of consecutive eigenvalue from the start that all satisfy $p < 0.01$ (p-values did not increase monotonically). For the evaluation with close relatives removed, each dataset was filtered with plink2 with option "--king-cutoff" with cutoff 0.02209709 ($= 2^{-11/2}$) for removing up to 4th degree relatives using KING-robust (*Manichaikul et al., 2010*), and MAF $< 0.01$ filter is reapplied (*Table 4*).

## Evaluation of performance

All approaches are evaluated using two complementary metrics: $\text{SRMSD}_p$ quantifies p-value uniformity, and $\text{AUC}_{\text{PR}}$ measures causal locus classification performance and reflects power while ranking miscalibrated models fairly. These measures are more robust alternatives to previous measures from the literature (Appendix 2), and are implemented in simtrait.

P-values for continuous test statistics have a uniform distribution when the null hypothesis holds, a crucial assumption for type I error and FDR control (*Storey, 2003*; *Storey and Tibshirani, 2003*). We use the Signed Root Mean Square Deviation ($\text{SRMSD}_p$) to measure the difference between the observed null p-value quantiles and the expected uniform quantiles:

$$\text{SRMSD}_p = \text{sgn}(u_{\text{median}} - p_{\text{median}}) \sqrt{\frac{1}{m_0} \sum_{i=1}^{m_0} \left(u_i - p_{(i)}\right)^2},$$

where $m_0 = m - m_1$ is the number of null (non-causal) loci, here $i$ indexes null loci only, $p_{(i)}$ is the $i$ th ordered null p-value, $u_i = (i - 0.5)/m_0$ is its expectation, $p_{\text{median}}$ is the median observed null p-value, $u_{\text{median}} = \frac{1}{2}$ is its expectation, and sgn is the sign function (1 if $u_{\text{median}} \geq p_{\text{median}}$, –1 otherwise). Thus, $\text{SRMSD}_p = 0$ corresponds to calibrated p-values, $\text{SRMSD}_p > 0$ indicate anti-conservative p-values, and $\text{SRMSD}_p < 0$ are conservative p-values. The maximum $\text{SRMSD}_p$ is achieved when all p-values are zero (the limit of anti-conservative p-values), which for infinite loci approaches

$$\text{SRMSD}_p \to \sqrt{\int_0^1 u^2 du} = \frac{1}{\sqrt{3}} \approx 0.577.$$

The same value with a negative sign occurs for all p-values of 1.

Precision and recall are standard performance measures for binary classifiers that do not require calibrated p-values (*Grau et al., 2015*). Given the total numbers of true positives (TP), false positives (FP) and false negatives (FN) at some threshold or parameter $t$, precision and recall are

$$\text{Precision}(t) = \frac{\text{TP}(t)}{\text{TP}(t) + \text{FP}(t)},$$

$$\text{Recall}(t) = \frac{\text{TP}(t)}{\text{TP}(t) + \text{FN}(t)}.$$

Precision and Recall trace a curve as $t$ is varied, and the area under this curve is $\text{AUC}_{\text{PR}}$. We use the R package PRROC to integrate the correct non-linear piecewise function when interpolating between points. A model obtains the maximum $\text{AUC}_{\text{PR}} = 1$ if there is a $t$ that classifies all loci perfectly. In contrast, the worst models, which classify at random, have an expected precision ($= \text{AUC}_{\text{PR}}$) equal to the overall proportion of causal loci: $m_1/m$.

## Data and code availability

The data and code generated during this study are available on GitHub at https://github.com/OchoaLab/pca-assoc-paper (copy archived at *Ochoa, 2023*). The public subset of Human Origins is available on the Reich Lab website at https://reich.hms.harvard.edu/datasets; non-public samples have to be requested from David Reich. The WGS version of HGDP was downloaded from the Wellcome Sanger Institute FTP site at ftp://ngs.sanger.ac.uk/production/hgdp/hgdp_wgs.20190516/. The high-coverage version of the 1000 Genomes Project was downloaded from ftp://ftp.1000genomes.ebi.ac.uk/vol1/ftp/data_collections/1000G_2504_high_coverage/working/20190425_NYGC_GATK/.

## Web resources

plink2, https://www.cog-genomics.org/plink/2.0/ ; GCTA, https://yanglab.westlake.edu.cn/software/gcta/ ; Eigensoft, https://github.com/DReichLab/EIG ; bnpsd, https://cran.r-project.org/package=

bnpsd ; simfam, https://cran.r-project.org/package=simfam ; simtrait, https://cran.r-project.org/package=simtrait ; genio, https://cran.r-project.org/package=genio ; popkin, https://cran.r-project.org/package=popkin ; ape, https://cran.r-project.org/package=ape ; nnls, https://cran.r-project.org/package=nnls ; PRROC, https://cran.r-project.org/package=PRROC ; BEDMatrix, https://cran.r-project.org/package=BEDMatrix.

## Acknowledgements

Thanks to Tiffany Tu, Ratchanon Pornmongkolsuk, and Zhuoran Hou for feedback on this article. This work was funded in part by the Duke University School of Medicine Whitehead Scholars Program, a gift from the Whitehead Charitable Foundation. The 1000 Genomes data were generated at the New York Genome Center with funds provided by NHGRI Grant 3UM1HG008901-03S1.

## Additional information

### Competing interests

Yiqi Yao: is affiliated with BenHealth Consulting. The author has no financial interests to declare. The other author declares that no competing interests exist.

### Funding

| Funder | Grant reference number | Author |
| --- | --- | --- |
| Whitehead Foundation | | Alejandro Ochoa |

The funders had no role in study design, data collection and interpretation, or the decision to submit the work for publication.

### Author contributions

Yiqi Yao, Software, Formal analysis, Investigation, Visualization, Writing – original draft, Writing – review and editing; Alejandro Ochoa, Conceptualization, Resources, Data curation, Software, Formal analysis, Supervision, Funding acquisition, Validation, Investigation, Visualization, Methodology, Writing – original draft, Project administration, Writing – review and editing

### Author ORCIDs

Alejandro Ochoa  http://orcid.org/0000-0003-4928-3403

### Decision letter and Author response

Decision letter https://doi.org/10.7554/eLife.79238.sa1
Author response https://doi.org/10.7554/eLife.79238.sa2

## Additional files

### Supplementary files

• MDAR checklist

### Data availability

The current manuscript is a computational study, so no data have been generated for this manuscript. Code is available at https://github.com/OchoaLab/pca-assoc-paper (copy archived at *Ochoa, 2023*).

The following previously published datasets were used:

| Author(s) | Year | Dataset title | Dataset URL | Database and Identifier |
|---|---|---|---|---|
| Fairley S | 2020 | 1000 Genomes Project, high-coverage version | ftp://ftp.1000genomes.ebi.ac.uk/vol1/ftp/data_collections/1000G_2504_high_coverage/working/20190425_NYGC_GATK/ | International Genome Sample Resource, NYGC_GATK/ |
| Bergstrom A | 2020 | Human Genome Diversity Panel, whole-genome sequencing version | ftp://ngs.sanger.ac.uk/production/hgdp/hgdp_wgs.20190516/ | Wellcome Sanger Institute, wgs.20190516/ |
| Lazaridis I | 2016 | Human Origins | https://reich.hms.harvard.edu/datasets | David Reich Lab, datasets |

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

## Appendix 1

### Fitting ancestral allele frequency distribution to real data

We calculated $\hat{p}_i^T$ distributions of each real dataset. However, population structure increases the variance of these sample $\hat{p}_i^T$ relative to the true $p_i^T$ (*Ochoa and Storey, 2021*). We present a new algorithm for constructing a new distribution based on the input data but with the lower variance of the true ancestral distribution. Suppose the $p_i^T$ distribution over loci $i$ satisfies $\mathrm{E}\left[p_i^T\right] = \frac{1}{2}$ and $\mathrm{Var}\left(p_i^T\right) = V^T$. The sample allele frequency $\hat{p}_i^T$, conditioned on $p_i^T$, satisfies

$$\mathrm{E}\left[\hat{p}_i^T\middle|p_i^T\right] = p_i^T, \qquad \mathrm{Var}\left(\hat{p}_i^T\middle|p_i^T\right) = p_i^T\left(1 - p_i^T\right)\bar{\varphi}^T,$$

where $\bar{\varphi}^T = \frac{1}{n^2}\sum_{j=1}^{n}\sum_{k=1}^{n}\varphi_{jk}^T$ is the mean kinship over all individual (*Ochoa and Storey, 2021*). The unconditional moments of $\hat{p}_i^T$ follow from the laws of total expectation and variance: $\mathrm{E}\left[\hat{p}_i^T\right] = \frac{1}{2}$ and

$$W^T = \mathrm{Var}\left(\hat{p}_i^T\right) = \bar{\varphi}^T\frac{1}{4} + \left(1 - \bar{\varphi}^T\right)V^T.$$

Since $V^T \leq \frac{1}{4}$ and $\bar{\varphi}^T \geq 0$, then $W^T \geq V^T$. Thus, the goal is to construct a new distribution with the original, lower variance of

$$V^T = \frac{W^T - \frac{1}{4}\bar{\varphi}^T}{1 - \bar{\varphi}^T}. \tag{9}$$

We use the unbiased estimator $\hat{W}^T = \frac{1}{m}\sum_{i=1}^{m}\left(\hat{p}_i^T - \frac{1}{2}\right)^2$, while $\bar{\varphi}^T$ is calculated from the tree parameters: the subpopulation coancestry matrix (*Equation 7*), expanded from subpopulations to individuals, the diagonal converted to kinship (reversing *Equation 5*), and the matrix averaged. However, since our model ignores the MAF filters imposed in our simulations, $\bar{\varphi}^T$ was adjusted. For Human Origins the true model $\bar{\varphi}^T$ of 0.143 was used. For 1000 Genomes and HGDP the true $\bar{\varphi}^T$ are 0.126 and 0.124, respectively, but 0.4 for both produced a better fit.

Lastly, we construct new allele frequencies,

$$p^* = w\hat{p}_i^T + (1 - w)q,$$

by a weighted average of $\hat{p}_i^T$ and $q \in (0, 1)$ drawn independently from a different distribution. $\mathrm{E}[q] = \frac{1}{2}$ is required to have $\mathrm{E}\left[p^*\right] = \frac{1}{2}$. The resulting variance is

$$\mathrm{Var}(p^*) = w^2 W^T + (1 - w)^2 \mathrm{Var}(q),$$

which we equate to the desired $V^T$ (*Equation 9*) and solve for $w$. For simplicity, we also set $\mathrm{Var}(q) = V^T$, which is achieved with:

$$q \sim \mathrm{Beta}\left(\frac{1}{2}\left(\frac{1}{4V^T} - 1\right), \frac{1}{2}\left(\frac{1}{4V^T} - 1\right)\right).$$

Although $w = 0$ yields $\mathrm{Var}(p^*) = V^T$, we use the second root of the quadratic equation to use $\hat{p}_i^T$:

$$w = \frac{2V^T}{W^T + V^T}.$$

# Appendix 2

## Comparisons between $\mathrm{SRMSD}_p$, $\mathrm{AUC_{PR}}$, and evaluation measures from the literature

### 2.1 The inflation factor $\lambda$

Test statistic inflation has been used to measure model calibration (**Astle and Balding, 2009**; **Price et al., 2010**). The inflation factor $\lambda$ is defined as the median $\chi^2$ association statistic divided by theoretical median under the null hypothesis (**Devlin and Roeder, 1999**). To compare p-values from non-$\chi^2$ tests (such as t-statistics), $\lambda$ can be calculated from p-values using

$$\lambda = \frac{F^{-1}\left(1 - p_{\mathrm{median}}\right)}{F^{-1}\left(1 - u_{\mathrm{median}}\right)},$$

where $p_{\mathrm{median}}$ is the median observed p-value (including causal loci), $u_{\mathrm{median}} = \frac{1}{2}$ is its null expectation, and $F$ is the $\chi^2$ cumulative density function ($F^{-1}$ is the quantile function).

To compare $\lambda$ and $\mathrm{SRMSD}_p$ directly, for simplicity assume that all p-values are null. In this case, calibrated p-values give $\lambda = 1$ and $\mathrm{SRMSD}_p = 0$. However, non-uniform p-values with the expected median, such as from genomic control (**Devlin and Roeder, 1999**), result in $\lambda = 1$, but $\mathrm{SRMSD}_p \neq 0$ except for uniform p-values, a key flaw of $\lambda$ that $\mathrm{SRMSD}_p$ overcomes. Inflated statistics (anti-conservative p-values) give $\lambda > 1$ and $\mathrm{SRMSD}_p > 0$. Deflated statistics (conservative p-values) give $\lambda < 1$ and $\mathrm{SRMSD}_p < 0$. Thus, $\lambda \neq 1$ always implies $\mathrm{SRMSD}_p \neq 0$ (where $\lambda - 1$ and $\mathrm{SRMSD}_p$ have the same sign), but not the other way around. Overall, $\lambda$ depends only on the median p-value, while $\mathrm{SRMSD}_p$ uses the complete distribution. However, $\mathrm{SRMSD}_p$ requires knowing which loci are null, so unlike $\lambda$ it is only applicable to simulated traits.

### 2.2 Empirical comparison of $\mathrm{SRMSD}_p$ and $\lambda$

There is a near one-to-one correspondence between $\lambda$ and $\mathrm{SRMSD}_p$ in our data (**Figure 2—figure supplement 1**). PCA tended to be inflated ($\lambda > 1$ and $\mathrm{SRMSD}_p > 0$) whereas LMM tended to be deflated ($\lambda < 1$ and $\mathrm{SRMSD}_p < 0$), otherwise the data for both models fall on the same contiguous curve. We fit a sigmoidal function to this data,

$$\mathrm{SRMSD}_p(\lambda) = a\frac{\lambda^b - 1}{\lambda^b + 1}, \tag{10}$$

which for $a, b > 0$ satisfies $\mathrm{SRMSD}_p(\lambda = 1) = 0$ and reflects $\log(\lambda)$ about zero ($\lambda = 1$):

$$\mathrm{SRMSD}_p(\log(\lambda) = -x) = -\mathrm{SRMSD}_p(\log(\lambda) = x).$$

We fit this model to $\lambda > 1$ only since it was less noisy and of greater interest, and obtained the curve shown in **Figure 2—figure supplement 1** with $a = 0.564$ and $b = 0.619$. The value $\lambda = 1.05$, a common threshold for benign inflation (**Price et al., 2010**), corresponds to $\mathrm{SRMSD}_p = 0.0085$ according to **Equation 10**. Conversely, $\mathrm{SRMSD}_p = 0.01$, serving as a simpler rule of thumb, corresponds to $\lambda = 1.06$.

### 2.3 Type I error rate

The type I error rate is the proportion of null p-values with $p \leq t$. Calibrated p-values have type I error rate near $t$, which may be evaluated with a binomial test. This measure may give different results for different $t$, for example be significantly miscalibrated only for large $t$ (due to lack of power for smaller $t$), and it requires large simulations to estimate well as it depends on the tail of the distribution. In contrast, $\mathrm{SRMSD}_p$ uses the entire distribution so it is easier to estimate, $\mathrm{SRMSD}_p = 0$ guarantees calibrated type I error rates at all $t$, while large $|\mathrm{SRMSD}_p|$ indicates incorrect type I errors for a range of $t$. Empirically, we find the expected agreement and monotonic relationship between $\mathrm{SRMSD}_p$ and type I error rate (**Figure 2—figure supplement 2**).

### 2.4 Statistical power and comparison to $\mathrm{AUC_{PR}}$

Power is the probability that a test is declared significant when the alternative hypothesis $H_1$ holds. At a p-value threshold $t$, power equals

$$F(t) = \Pr(p < t | H_1).$$

$F(t)$ is a cumulative function, so it is monotonically increasing and has an inverse. Like type I error control, power may rank models differently depending on $t$, and it is also harder to estimate than $\text{AUC}_{\text{PR}}$ because power depends on the tail of the distribution.

Power is not meaningful when p-values are not calibrated. To establish a clear connection to $\text{AUC}_{\text{PR}}$, assume calibrated (uniform) null p-values: $\Pr(p < t | H_0) = t$. TPs, FPs, and FNs at $t$ are

$$
\begin{aligned}
\text{TP}(t) &= m\pi_1 F(t), \\
\text{FP}(t) &= m\pi_0 t, \\
\text{FN}(t) &= m\pi_1(1 - F(t)),
\end{aligned}
$$

where $\pi_0 = \Pr(H_0)$ is the proportion of null cases and $\pi_1 = 1 - \pi_0$ of alternative cases. Therefore,

$$
\begin{aligned}
\text{Precision}(t) &= \frac{\pi_1 F(t)}{\pi_1 F(t) + \pi_0 t}, \\
\text{Recall}(t) &= F(t).
\end{aligned}
$$

Noting that $t = F^{-1}(\text{Recall})$, precision can be written as a function of recall, the power function, and constants:

$$\text{Precision}(\text{Recall}) = \frac{\pi_1 \text{Recall}}{\pi_1 \text{Recall} + \pi_0 F^{-1}(\text{Recall})}.$$

This last form leads most clearly to $\text{AUC}_{\text{PR}} = \int_0^1 \text{Precision}(\text{Recall}) d\text{Recall}$ .

Lastly, consider a simple yet common case in which model $A$ is uniformly more powerful than model $B : F_A(t) > F_B(t)$ for every $t$. Therefore $F_A^{-1}(\text{Recall}) < F_B^{-1}(\text{Recall})$ for every recall value. This ensures that the precision of $A$ is greater than that of $B$ at every recall value, so $\text{AUC}_{\text{PR}}$ is greater for $A$ than $B$. Thus, $\text{AUC}_{\text{PR}}$ ranks calibrated models according to power.

Empirically, we find the predicted positive correlation between $\text{AUC}_{\text{PR}}$ and calibrated power (*Figure 2—figure supplement 3*). The correlation is clear when considered separately per dataset, but the slope varies per dataset, which is expected because the proportion of alternative cases $\pi_1$ varies per dataset.

