## [Editor Report]

This is an important paper that presents compelling arguments (based on simulation and comprehensively reviewed background theory) that Linear Mixed Models generally should perform better at correcting for genetic and environmental confounding in GWAS than more commonly used Principal Components methods.

---

## [Decision Letter]

**Decision letter after peer review:**

Thank you for submitting your article "Limitations of principal components in quantitative genetic association models for human studies" for consideration by *eLife*. Your article has been reviewed by 4 peer reviewers, including Magnus Nordborg as the Reviewing Editor and Reviewer #1, and the evaluation has been overseen by Detlef Weigel as the Senior Editor.

The reviewers have discussed their reviews with one another, and the Reviewing Editor has drafted this to help you prepare a revised submission. Individual reviews are also included as they are generally helpful. As you will see, there is strong support for your work, but it is clear that improvements are in order to justify your general claims.

Essential revisions:

1) You must at least discuss that the scenarios you test are in a sense as unrealistic as the (much more limited) simulations previously run, both in terms of sample size and population structure. Actual human GWAS are run "within populations" to minimize environmental confounding (more on this below) as much long-range LD and involve much larger sample sizes. Does this affect your conclusions? Related to this, the historical context for several methods you cite is not given.

2) Practical considerations for why PCs were/are used are not discussed either. Unbalanced designs, meta-studies, sample sizes.

3) You are not discussing environmental confounding, which is not only likely more of a problem in human GWAS than any of the other things discussed here, but could potentially justify a combination of LMM and PCs. Selection could have a similar effect, as discussed in the original LMM paper by Yu et al. (2006). Ideally, this should be simulated, but it certainly must be discussed, and the conclusions must take this into account.

4) You are similarly not addressing the issue of rare alleles and heterogeneity, which is a major preoccupation in human genetics. How does genetic architecture affect your conclusions? If you could address this, it would be a very nice (and timely) contribution (whereas the manuscript as currently written feels like it's fighting a 10-year old battle).

5) Finally, the description of the theory is often hard to follow, relies heavily on a particular model (ancient allele frequencies), and appears to be wrong in a few specific cases (see comments below). The results table is likewise inscrutable, and it is difficult to relate your simulations to real-life scenarios.

*Reviewer #1 (Recommendations for the authors):*

At the very least you need to comment on the practicability and relevance issues noted above.

I would also recommend adjusting the writing to make your (otherwise lucid) review of the theory more accessible to a broader audience. Not only do you assume considerable knowledge of mathematical statistics, but you are also deeply rooted in a classical quantitative genetics framework. Terms like "inbreeding edges" are likely to befuddle all but a tiny fraction of humanity!

*Reviewer #2 (Recommendations for the authors):*

1. Environmental effects are known to correlate with population structure. A silly example of this is chop-stick skills, which obviously correlate with broad levels of East-Asian ancestry, e.g. in a global population sample it would likely correlate with the first two top PCs, and additional PCs would not explain more. Hence, for such a trait the generative model could really just be the top PCs, in which case including top PCs as covariates would be optimal. An LMM would in this example probably try to account for a much more complicated structure in addition to the top PCs. Hence, in summary, I would appreciate some simulations where you have environmental contributions that are perhaps strongly correlated with individual PCs, as I believe such scenarios may exist in real data analyses.

2. I really appreciate the detailed simulations in this study, but perhaps it would make sense to simulate traits given UKB genotypes? Also, and perhaps more importantly, how about also evaluating the two approaches on UKB data, to see if you really find more hits using LMMs? (similar to Loh et al., NG 2018)

3. You note that PCs are the top eigenvectors of the kinship matrix. This is usually true, but not always as when deriving PCs one should ideally apply some LD adjustment to avoid PCs capturing long-range LD regions, which can otherwise reduce power to detect variants in long-range regions. See e.g. Patterson et al. (PLoS Genet 2006) or Privé et al. (Bioinformatics 2020).

4. Following up on the previous comment, I wonder if the conclusions in this paper change if PCs are derived the way they often are, i.e. excluding long-range LD regions and/or using some LD adjustments.

5. Some of the LMMs cited are not classical LMMs, in the sense that they do not assume an infinitesimal genetic architecture. E.g. BOLT-LMM (Loh et al., Nat Genet 2015) assumes a mixture of two Gaussians as a prior for the effects, which they show can improve power further. Mbatchou et al. (Nat Genet 2021) also use blockwise ridge regression (infinitesimal model), which is effectively a more flexible prior. Using these, the relationship between the PCs and the model becomes yet more complicated. Similarly, the GCTA mixed model GWAS uses a LOCO approach to improve power, which I believe means that a slightly different kinship is used for each chromosome.

6. Your derivation assumes that there are some ancestral allele frequencies underlying the true model, but that seems perhaps unnecessary to me because these allele frequencies only represent the "correct" weights for the variants, and we know that they are probably wrong anyway. Indeed alternative and likely better weightings can be used, see Speed and Balding (AJHG 2012, NGR 2015, NG 2019). If you instead assume the sample frequencies are the right frequencies, then all the math becomes simple by standardizing the variants.

7. Another publication that also provides some similar theory on the relationship between PCA and mixed models is Janss et al., (Genetics 2012). (Interestingly, although their work is very nice their PCs did however have major long-range LD issues.)

8. The authors mention that LMMs are not well suited for unbalanced samples, e.g. where the case-control ratio in the sample is <5%, and cite Zhou et al., Nat Genet 2018 as a solution. However, before Zhou et al. there were no computationally efficient generalized mixed models capable of being able to be applied to UKB sample sizes, which could help explain why LMMs haven't been adopted as quickly as one might have expected.

9. I believe there is some concern regarding meta-analyzing LMM GWAS summary statistics, e.g. whether one should use the actual effective sample sizes or the effective sample sizes (which I believe is the best approach). I believe this is probably the main reason why LMMs are not as widely used as we would like. I would appreciate some discussion or reflection on this.

*Reviewer #3 (Recommendations for the authors):*

The presentation of the results in Table 3 is nearly incomprehensible.

The authors are incorrect about no other troublesome cases on line 48. The differential variance between populations can induce bias (see papers from Xihong Lin).

The author's discussion of the benefits of including sample PCs in LMM (eg 468-472 on p28) is misinformed. In particular, it is a critical step for methods like BOLT.

The authors say "low-dimensional" when they mean "low-rank".

The Sul and Eskin paper does not result in a tie. It states that creating a second kinship matrix from the PCs can address the issue presented in Price et al. entirely with LMMs.

It would also be interesting to know to what extent the singular value distribution matters versus the rank? The authors have some discussion of this in 5b but it is not well developed.

*Reviewer #4 (Recommendations for the authors):*

I have the following comments for consideration:

Historical context: The overall reasoning for controlling population stratification and relative kinship for different types of populations was given in [26]. It would be helpful to revise the introduction part to focus more on the original research papers and their reasonings: [5] (PCA), [9] (Q), [26] (LMM, Q in MM), and [16] (PC in LMM and actual association study). The later studies with LMM from the groups behind GCTA should be mentioned as the (emerging/emerged) trend, in addition to those comparison studies.

A couple of relevant research may be examined and discussed, in terms of different types of populations, determining the number of PCs to be included in the LMM, model comparison, and modeling fitting (redundancy). https://doi.org/10.1534/genetics.108.098863 https://doi.org/10.1038/hdy.2010.11

One question is the justification for examining a single heritability of 0.8. This level is generally regarded very high. It would be more convincing to see the results when h2 = 0.3-0.6, particularly if this is set for a large population with different subgroups.

The choice of m1=n/10 is not adequately justified. Even though we typically assume that there are many loci and the detection power increases with larger sample sizes, the actual detected numbers of loci in empirical studies are lower than that. With the current set m1 (Table 2), what were the power values, and are they close to the empirical studies?

"The largest limitation of our work is that we only considered quantitative traits". This may be expanded to include that with the overall simulation scheme, you assume the causal variants are functioning across different groups of a large population (2.2.5 Trait Simulation)? Real data with measured traits may be different. This can also be different for different types of complex diseases. Some clear context information should be given at the beginning too.

Among these simulated population and real data, have you examined whether a small number of PCs are indeed significant to explain some trait variation? I think this was the original thinking of applying PCs to correct the population stratification.

L46 and L487. Assume the low-dimension part of the relatedness matters in terms of trait differences among groups, which need to be adjusted.

L53-57. This is not an accurate summary of the relevant papers. Earlier papers set up the general LMM framework with different components that may be included or included and individual components that can have varied forms or reported the first actual association study with the LMM framework. These two are earlier papers that (re-)introduce LMM to the association studies, and markers were used for kinship, structure, and PCA.

L64-70. Redundancy is not an issue since the objective is to control false positives using two types of covariates.

L166. Large "and"(?) Family.

L282. Theoretical and empirical evidence of these two needs to be provided.

Table 3. Not clear about the asterisk.

---

## [Author Response]

Essential revisions:1) You must at least discuss that the scenarios you test are in a sense as unrealistic as the (much more limited) simulations previously run, both in terms of sample size and population structure. Actual human GWAS are run "within populations" to minimize environmental confounding (more on this below) as much long-range LD and involve much larger sample sizes. Does this affect your conclusions? Related to this, the historical context for several methods you cite is not given.

We added to our discussion comments on sample size, which is smaller than many of the largest modern studies. However, we do not expect our key conclusion to change for larger sample sizes, since cryptic family relatedness is not only present there but it is expected to be even more abundant compared to smaller sample sizes.

Many modern GWAS are multiethnic or include admixed individuals, so this is not an artificial setting that has no bearing on real analyses. We now cite 21 recent papers on this topic to back up this assertion. We focused on these more extreme cases as this is where challenges to PCA and LMM performance are most expected. We agree that environmental confounding is an important issue for multiethnic studies, and have now included numerous simulations with considerable environment effects.

Our simulated genotypes do not have LD, and our real data was pruned to exclude short range LD in order to simplify evaluations. The reviewers point out several variants to PC and kinship estimation that can improve handling of LD, and we now incorporate those in our discussion. We agree LD should be handled with appropriate estimators when present, but overall we do not think the presence of LD changes our key conclusion, namely that LMM outperforms PCA and that this is mostly due to the presence of cryptic family relatedness.

For clarity, we opted to focus on describing the most common modern versions of PCA and LMM which are tested, as opposed to numerous older variants we did not test here, which modeled population and family structure in somewhat different ways. We provide historical context strictly as needed to avoid confusion and keep our already lengthy work as brief as possible.

2) Practical considerations for why PCs were/are used are not discussed either. Unbalanced designs, meta-studies, sample sizes.

We added a discussion paragraph to sample sizes and related questions. Indeed, PCA scales better than classic LMMs with sample size, although many recent LMMs are now scaling better, although some do so by changing the inference model somewhat (by taking new shortcuts, for example), so those new models deserve further study. However, cryptic family relatedness is expected to become more abundant at larger sample sizes, so PCA should perform worse there.

We also briefly discuss unbalanced case-control designs. However, quantitative trait studies do not typically exhibit unbalanced designs, which is the sole focus of this work, so this interesting question is out of scope.

Meta-analysis does not give either PCA or LMM an advantage, as it can be applied to summary statistics from either or even a combination of models.

3) You are not discussing environmental confounding, which is not only likely more of a problem in human GWAS than any of the other things discussed here, but could potentially justify a combination of LMM and PCs. Selection could have a similar effect, as discussed in the original LMM paper by Yu et al. (2006). Ideally, this should be simulated, but it certainly must be discussed, and the conclusions must take this into account.

We added numerous simulations with large environment effects. Although an LMM with PCs can occasionally significantly outperform an LMM without PCs, in these cases the difference in performance tended to be small, and including environment variables as fixed effects in our regression improves performance much more than using PCs. Therefore, we are hesitant to recommend always including PCs in an LMM since this tends to be a poor solution to the problem of environmental confounding, and more explicit modeling of environment effects is instead recommended.

4) You are similarly not addressing the issue of rare alleles and heterogeneity, which is a major preoccupation in human genetics. How does genetic architecture affect your conclusions? If you could address this, it would be a very nice (and timely) contribution (whereas the manuscript as currently written feels like it's fighting a 10-year old battle).

We now include rare variants in our discussion. In our evaluations we had to ignore rare variants because at our small sample sizes we have no power to detect association at these variants with low counts (and p-values were often miscalibrated in those cases, a well known issue). However, we agree that modeling rare variants is important in biobank-scale analyses, and their covariance structure is different than for common variants (Zaidi and Mathieson, 2020), which raises very interesting challenges we hope to tackle in future work.

By heterogeneity, one reviewer clarifies below to mean effect size heterogeneity across populations, which is the focus of much recent work. However, this questions is not relevant to our LMM vs PCA comparisons, as neither of these basic models is equipped to model effect size heterogeneity, neither is expected a priori to perform better than the other, and both models can be extended to model ancestry-specific effects in the way it is done in TRACTOR, for example.

5) Finally, the description of the theory is often hard to follow, relies heavily on a particular model (ancient allele frequencies), and appears to be wrong in a few specific cases (see comments below). The results table is likewise inscrutable, and it is difficult to relate your simulations to real-life scenarios.

We updated the descriptions of our theory as suggested in the more specific comments below, and there we also address the particulars of our model. We improved Table 3 with the main statistical evaluations.

Reviewer #1 (Recommendations for the authors):At the very least you need to comment on the practicability and relevance issues noted above.I would also recommend adjusting the writing to make your (otherwise lucid) review of the theory more accessible to a broader audience. Not only do you assume considerable knowledge of mathematical statistics, but you are also deeply rooted in a classical quantitative genetics framework. Terms like "inbreeding edges" are likely to befuddle all but a tiny fraction of humanity!

We clarified the concepts highlighted and additionally edited based on feedback from additional members of our lab who reviewed the revised manuscript.

Reviewer #2 (Recommendations for the authors):1. Environmental effects are known to correlate with population structure. A silly example of this is chop-stick skills, which obviously correlate with broad levels of East-Asian ancestry, e.g. in a global population sample it would likely correlate with the first two top PCs, and additional PCs would not explain more. Hence, for such a trait the generative model could really just be the top PCs, in which case including top PCs as covariates would be optimal. An LMM would in this example probably try to account for a much more complicated structure in addition to the top PCs. Hence, in summary, I would appreciate some simulations where you have environmental contributions that are perhaps strongly correlated with individual PCs, as I believe such scenarios may exist in real data analyses.

We added numerous simulations with random environment effects generated from the population labels given by the three real datasets (and with labels that group individuals by geographic coordinates for the simulated datasets). These environment effects are strongly correlated with ancestry and are also low-dimensional, which could give PCA an advantage based on those properties as you and many others expect. By necessity, the environment simulations have a lower heritability, to accommodate for large environment effects. Thus, for comparison, we also simulated numerous low heritability traits without environment, whose results are comparable to the original high heritability simulations. Surprisingly, we find that LMMs fit environment effects nearly as well as PCA, since both increase their AUCs substantially compared to the low heritability simulations (Figure 8 Figure supp 1). We do find that with environment, occasionally, an LMM with PCs slightly but significantly outperforms an LMM without PCs, and in other cases PCA similarly outperforms LMM. But we do want to emphasize that the effect is often not there, and that in general LMM without PCs performs nearly as well as the best competing method, and significant differences are invariably small. Lastly, we included an LMM with the true group labels as fixed covariates, and in most cases that was the best method and its AUCs were greater by considerable amounts. Overall, we do see this potential advantage to including PCs, but it is in no way a reliable solution, and it is no replacement for more direct modeling of environment effects.

2. I really appreciate the detailed simulations in this study, but perhaps it would make sense to simulate traits given UKB genotypes? Also, and perhaps more importantly, how about also evaluating the two approaches on UKB data, to see if you really find more hits using LMMs? (similar to Loh et al., NG 2018)

We are interested in this question of larger datasets, but it is out of scope of the current work. One immediate concern is that the LMM used in our paper (GCTA) cannot be used in these larger datasets. GCTA is in a sense a classic LMM that in our evaluations performs as well as older methods (EMMAX, GEMMA) but is faster than those, so we have no concerns that either of those would have similar power if we were able to run them on UKB (but this is not possible). In contrast, the scalable LMM approaches applicable to the UKB (BOLT-LMM, SAIGE, REGENIE, and others) do so by making additional approximations that older LMMs do not use, which worry us as they may potentially reduce power and could confound the results. BOLT-LMM in particularly has performed poorly in our hands in simulations with *n* = 1000 individuals (unpublished results), so this is not a merely hypothetical concern. A fair and complete evaluation that considers all of these variables is an entire line of work separate from the current evaluation, but it is definitely the next step.

That said, we believe the main conclusion will not change on UKB or other large datasets. In particular, we expect cryptic family relatedness to remain a serious problem, and perhaps have a larger impact as the abundance of distant relatives only increases with sample size.

3. You note that PCs are the top eigenvectors of the kinship matrix. This is usually true, but not always as when deriving PCs one should ideally apply some LD adjustment to avoid PCs capturing long-range LD regions, which can otherwise reduce power to detect variants in long-range regions. See e.g. Patterson et al. (PLoS Genet 2006) or Privé et al. (Bioinformatics 2020).4. Following up on the previous comment, I wonder if the conclusions in this paper change if PCs are derived the way they often are, i.e. excluding long-range LD regions and/or using some LD adjustments.

We incorporated these variants of the PCA approach into our discussion, and in particular added the following conclusion: “These improved PCs remain inadequate models of family relatedness, so an LMM will continue to outperform them in that setting.”

5. Some of the LMMs cited are not classical LMMs, in the sense that they do not assume an infinitesimal genetic architecture. E.g. BOLT-LMM (Loh et al., Nat Genet 2015) assumes a mixture of two Gaussians as a prior for the effects, which they show can improve power further. Mbatchou et al. (Nat Genet 2021) also use blockwise ridge regression (infinitesimal model), which is effectively a more flexible prior. Using these, the relationship between the PCs and the model becomes yet more complicated. Similarly, the GCTA mixed model GWAS uses a LOCO approach to improve power, which I believe means that a slightly different kinship is used for each chromosome.

We incorporated the LOCO approach into our discussion, and added the following conclusion: “While LOCO kinship estimates vary for each chromosome, they continue to model family relatedness, thus maintaining their key advantage over PCA.”

We also incorporated BOLT-LMM and REGENIE into a different part of our discussion, and agreed with your conclusion: “Many of these newer LMMs also no longer follow the infinitesimal model of the basic LMM (Loh et al., 2015; Mbatchou et al., 2021), and employ novel approximations, which are features not evaluated in this work and worthy of future study.”

6. Your derivation assumes that there are some ancestral allele frequencies underlying the true model, but that seems perhaps unnecessary to me because these allele frequencies only represent the "correct" weights for the variants, and we know that they are probably wrong anyway. Indeed alternative and likely better weightings can be used, see Speed and Balding (AJHG 2012, NGR 2015, NG 2019). If you instead assume the sample frequencies are the right frequencies, then all the math becomes simple by standardizing the variants.

Our methods contained early on the following explanation: “However, this kinship estimator has a complex bias that differs for every individual pair, which arises due to the use of this estimated *p*ˆ*^T^_i_* (Ochoa and Storey, 2021, 2019). Nevertheless, in PCA and LMM these biased estimates perform as well as unbiased ones (Hou and Ochoa, 2023).”

To further explain, our recent work (Ochoa and Storey, 2021) has shown that sample allele frequencies are not the right frequencies, although that is a pervasive assumption, and in particular kinship estimates are biased when making this precise assumption. In that work we developed a new estimator, popkin, that is unbiased and overcomes the problem that the true ancestral allele frequencies are unknown, and estimates kinship without bias without having to estimate such allele frequencies at all. We use popkin to visualize population structure in Figure 1. and to calculate eigenvalues later. However, our followup work found that PCA and LMM work just fine with standard biased kinship estimates, because the intercept compensates (in a complicated way) for the bias in PCs or random effects (Hou and Ochoa, 2023). Nevertheless, it is very important to distinguish between true and estimated allele frequencies, as biases creep in when such distinctions are ignored, and regardless of whether this is ultimately the correct model or not (no model is completely correct, of course).

Since Speed and Balding (2012) is relevant to a new related discussion about PCA and LMM variants, all of which attempt to improve LD modeling, we added this paper to that discussion as well.

7. Another publication that also provides some similar theory on the relationship between PCA and mixed models is Janss et al., (Genetics 2012). (Interestingly, although their work is very nice their PCs did however have major long-range LD issues.)

We incorporated Janss et al., (2012) in our paper, cited in introduction, methods, and discussion.

8. The authors mention that LMMs are not well suited for unbalanced samples, e.g. where the case-control ratio in the sample is <5%, and cite Zhou et al., Nat Genet 2018 as a solution. However, before Zhou et al. there were no computationally efficient generalized mixed models capable of being able to be applied to UKB sample sizes, which could help explain why LMMs haven't been adopted as quickly as one might have expected.

We agree, and added a discussion paragraph concerning the sample size limitations of the present work that echo these points.

9. I believe there is some concern regarding meta-analyzing LMM GWAS summary statistics, e.g. whether one should use the actual effective sample sizes or the effective sample sizes (which I believe is the best approach). I believe this is probably the main reason why LMMs are not as widely used as we would like. I would appreciate some discussion or reflection on this.

We agree these are important questions to be analyzed in future work, but meta-analysis is a very different kind of model that falls outside the scope of this present work.

Reviewer #3 (Recommendations for the authors):The presentation of the results in Table 3 is nearly incomprehensible.

We modified Table 3 considerably for greater clarity. We simplified the statistical tests to center on the hypothesis that LMM with no PCs is better than all other cases (before we instead sought to identify smaller numbers of PCs that performed the same, statistically, as the best number of PCs, but that was both confusing and did not directly address our central hypothesis). For additional clarity we now include all p-values. Now we use a Bonferroni threshold to declare significance (the original used a fixed p-value threshold).

The authors are incorrect about no other troublesome cases on line 48. The differential variance between populations can induce bias (see papers from Xihong Lin).

We apologize, we were unable to identify the precise papers from Xihong Lin you are referring to, using your keywords of PCA and differential variance between populations. However, we stand by our comment, in that numerous papers have proposed various explanations for poor PCA performance that have not stood the test of time, and family relatedness is the only explanation that recurs and that numerous papers/authors agree upon.

The author's discussion of the benefits of including sample PCs in LMM (eg 468-472 on p28) is misinformed. In particular, it is a critical step for methods like BOLT.

We added the following to the discussion (appears a few paragraphs after the lines cited above): “A different benefit for including PCs were recently reported for BOLT-LMM, which does not result in greater power but rather in reduced runtime, a property that may be specific to its use of scalable algorithms such as conjugate gradient and variational Bayes (Loh et al., 2018).”

To further clarify, the BOLT-LMM authors make no claim that there are gains in performance (which in our paper refers exclusively to calibration and power) due to inclusion of PCs, it is solely a runtime advantage. Further, this appears to be a property of the specific algorithm used by BOLT-LMM, and not a general property of LMMs. Thus, our original statement stands.

The authors say "low-dimensional" when they mean "low-rank".

Low-rank is more correct mathematically, so we included a clarification in the first mention of “low-dimensional” relatedness. We also replaced some mentions of dimension with “matrix rank” when that is more precise, and in other cases added the adjective “model dimension” when that was appropriate. We also entirely replaced the vague term “dimensionality” with more precise terms, which varied depending on context. However, we thought “low-rank” may be too technical to use exclusively, so we did not fully excise the more intuitive “low-dimensional” expression from our work.

The Sul and Eskin paper does not result in a tie. It states that creating a second kinship matrix from the PCs can address the issue presented in Price et al. entirely with LMMs.

You are correct. However, we sought to indicate whether a standard LMM (with a single, standard kinship matrix) outperformed PCA, and in that regard it was a tie. LMMs with multiple kinship matrices are not commonly used and are not the subject of our investigation.

It would also be interesting to know to what extent the singular value distribution matters versus the rank? The authors have some discussion of this in 5b but it is not well developed.

We indeed consider this in what is now Figure 6 Figure supp 1, which shows the cumulative distribution of eigenvalues. However, that figure shows that none of the datasets stand out in having exceptional distributions except for the admixed family simulation. Since these distributions do not clearly separate the cases where PCA performed poorly from the rest, we decided that a brief mention of this analysis in the main text was enough to state that it was done but it was inconclusive. We think that spending additional time on this tangent is distracting from the stronger results we obtained from looking at the local kinship distributions.

Reviewer #4 (Recommendations for the authors):I have the following comments for consideration:Historical context: The overall reasoning for controlling population stratification and relative kinship for different types of populations was given in [26]. It would be helpful to revise the introduction part to focus more on the original research papers and their reasonings: [5] (PCA), [9] (Q), [26] (LMM, Q in MM), and [16] (PC in LMM and actual association study). The later studies with LMM from the groups behind GCTA should be mentioned as the (emerging/emerged) trend, in addition to those comparison studies.

For reference, these are the citations (as numbers may have changed in revision):

[5]: Price et al., (2006): PCA[9]: Pritchard et al., (2000): Q[16]: Zhao et al., (2007): PC in LMM and actual association[26]: Yu et al., (2006): LMM, Q in MM

Earlier draft versions of our introduction presented the historical context in more detail, similarly to what you are suggesting, but after much consideration we converged onto the current version, which we believe is appropriately more focused on the current approaches that were tested. In particular, our paper does not test the Q association model directly, and only tests one newer LMM that uses the standard kinship estimator (unlike the above early LMMs), so we thought spending more time explaining previous approaches that we are not testing can be distracting and confusing to readers. We thought the text flowed better the way it is written, not only primarily describing current approaches, but also describing PCA and LMM papers separately rather than chronologically (the timelines overlap considerably). We chose to focus on explaining the methods and their properties clearly instead. We do cover the conclusions of those early papers and highlight the differences between those and current approaches, but we do so strictly as needed (not chronologically) and as concisely as possible for clarity.

A couple of relevant research may be examined and discussed, in terms of different types of populations, determining the number of PCs to be included in the LMM, model comparison, and modeling fitting (redundancy). https://doi.org/10.1534/genetics.108.098863 https://doi.org/10.1038/hdy.2010.11

Zhu and Yu, 2009: Thank you for making us aware of this excellent evaluation, we included it in Table 1, and cited it in numerous locations in the introduction and discussion.

Sun et al., 2010: We now cite this paper in the discussion: “Additionally, PCA and LMM goodness of fit could be compared using the coefficient of determination generalized for LMMs (Sun et al., 2010).” Although we would have liked to try calculating these *R_LR_*^2^ in all of our simulations, for the baseline PCA and LMM models without loci, it comes at a considerable computational expense for LMMs, especially for the largest datasets, so we decided not to pursue it for this work. Furthermore, as variance components are estimated in GCTA using REML, we are also unsure if those estimates are appropriate for use in *R_LR_*^2^ (technically estimates must be ML, not REML).

One question is the justification for examining a single heritability of 0.8. This level is generally regarded very high. It would be more convincing to see the results when h2 = 0.3-0.6, particularly if this is set for a large population with different subgroups.

We repeated most of our simulations with a heritability of 0.3, and all of our conclusions hold.

The choice of m1=n/10 is not adequately justified. Even though we typically assume that there are many loci and the detection power increases with larger sample sizes, the actual detected numbers of loci in empirical studies are lower than that. With the current set m1 (Table 2), what were the power values, and are they close to the empirical studies?

Since we now consider varying heritability, we extended our heuristic formula to “*m*_1_ = round(*nh*^2^*/*8), which empirically balances power well with varying *n* and *h*^2^.” Note that for the original heritability of *h*^2^ = 0*.*8 the above formula reduces to the original one. This heuristic formula aims to maintain similar power (or AUCs) across studies with different parameters, and it works well enough empirically, though it is not exact and we did observe that AUCs vary somewhat between studies. More parameters surely matter for determining AUC.

We do not recover most of the causal loci we are simulating, reflected in the fact that AUCs are near 0.1 or 0.2 (this is, roughly speaking, close to the proportion of causal loci we are recovering confidently, which are clearly separated from the noise). We added a new figure that compares AUC to power, and find rough agreement between AUC and calibrated power calculated at an empirical type I error rate of 1e-4 (Figure 2 Figure supp 3). Therefore, our power at that level is roughly 0.1 to 0.2. In contrast, other empirical studies try to select parameters so that power is near 50%, although the p-value thresholds at which power is measured varies considerably in those previous studies. We think our simulation is more realistic in the sense that most causal loci are not recovered by GWAS, but their presence is important as they add to the background polygenic effect, so it makes sense to model a large number of causal loci even if we can only hope to recover a small proportion of them.

"The largest limitation of our work is that we only considered quantitative traits". This may be expanded to include that with the overall simulation scheme, you assume the causal variants are functioning across different groups of a large population (2.2.5 Trait Simulation)? Real data with measured traits may be different. This can also be different for different types of complex diseases. Some clear context information should be given at the beginning too.

By this we assume you mean that we did not simulate ancestry-specific effect size heterogeneity (in other words, our coefficients are the same for all individuals, of all ancestries). Although some work has suggested that effect sizes can vary across ancestries, other recent work shows that this is not common (K. Hou et al., 2023). However, this is not a limitation in the context of our paper, as neither the basic PCA or LMM are equipped to handle effect size heterogeneity, so adding such details to our simulation is not expected to benefit either approach. There are more recent approaches that extend that model effect size heterogeneity, such as TRACTOR, but they are not the focus of this evaluation. Additionally, both PCA and LMM can, in principle, be extended to model effect heterogeneity in exactly the same way as TRACTOR (which is a fixed effects model most like PCA).

Among these simulated population and real data, have you examined whether a small number of PCs are indeed significant to explain some trait variation? I think this was the original thinking of applying PCs to correct the population stratification.

We now performed these tests (Figure 6 Figure supp 2) and they always yielded small numbers of PCs that were significant: minimum 1 observed for the small sample size admixture simulation, most other cases had several significant PCs, maximum below 20 observed for the Admixed Family simulation and for Human Origins (both real and the tree simulated version). Overall they were roughly consistent with the numbers of PCs needed for the PCA method to maximize its performance in each of these datasets.

We introduced the following text in the section where we discuss the latent dimension of datasets and whether they could be used to predict when PCA would perform poorly (they don’t): “We also calculated the number of PCs that are significantly associated with the trait, and observed similar results, namely that while the family simulation has more significant PCs than the non-family admixture simulations, the real datasets and their tree simulated counterparts have similar numbers of significant PCs (Figure 6 Figure supp 2).”

L46 and L487. Assume the low-dimension part of the relatedness matters in terms of trait differences among groups, which need to be adjusted.

It is unnecessary to state that PCA assumes that the effect on the trait is also low-dimensional. Under non-zero heritability, high-dimensional relatedness (family structure) always affects the trait. It is impossible for the effect to be restricted to a low-dimensional component, because close relatives will share an outsize proportion of variants (whether they are common or rare in different groups) and thus have much more similar traits than random people (in the same group or otherwise).

L53-57. This is not an accurate summary of the relevant papers. Earlier papers set up the general LMM framework with different components that may be included or included and individual components that can have varied forms or reported the first actual association study with the LMM framework. These two are earlier papers that (re-)introduce LMM to the association studies, and markers were used for kinship, structure, and PCA.

We reworded the first sentence to: “Early LMMs used kinship matrices estimated from known pedigrees or using methods that captured recent relatedness only, and modeled population structure (ancestry) as fixed effects (Yu et al., 2006; Zhao et al., 2007; Zhu et al., 2009).” The intent of that sentence was not to describe early LMMs entirely, or as abstract frameworks, but to point out that the specific kinship estimates used then in practice were different than the ones commonly used now, though we agree the original version was misleading.

L64-70. Redundancy is not an issue since the objective is to control false positives using two types of covariates.

We agree, redundancy does not often cause problems in practice, but without a clear benefit it’s not worth the clear risk of loss of power (which we demonstrate for large numbers of PCs) and increased computational burden.

L166. Large "and"(?) Family.

We reworded to say “Both Large and Family simulations …”.

L282. Theoretical and empirical evidence of these two needs to be provided.

We previously provided theoretical justification for our SRMSDp and AUC metrics in Appendix B, showing that SRMSDp reflects null p-value calibration and agrees with the inflation factor and type I error rate, while AUC corresponds to calibrated power. We also provided Figure 2 Figure supp 1, which showed an excellent monotonic correspondence between SRMSDp and the inflation factor. We added two supplementary figures that provide additional empirical justification. Figure 2 Figure supp 2 shows that SRMSDp and type I error are monotonically related, and that they agree as expected for calibrated models. Figure 2 Figure supp 3 shows a positive correlation between AUC and calibrated power, which holds well separately per dataset, although the slope of this relationship varies between datasets because they have different proportions of loci where the alternative hypothesis holds.

Table 3. Not clear about the asterisk.

This table has been reorganized at the request of another reviewer. The asterisk denoted that model with that number of PCs was calibrated (determined by whether the mean absolute SRMSDp was below 0.01). In the new table there is instead a column that denotes if each model is calibrated as a boolean (True or False).